# Structure of E3 ligase E6AP with a proteasome-binding site provided by substrate receptor hRpn10

Gwen R. Buel [1,6], Xiang Chen [1,6✉], Raj Chari[2], Maura J. O'Neill[3], Danielle L. Ebelle[1], Conor Jenkins[3], Vinidhra Sridharan[1], Sergey G. Tarasov[4], Nadya I. Tarasova[5], Thorkell Andresson[3] & Kylie J. Walters[1✉]

Regulated proteolysis by proteasomes involves ~800 enzymes for substrate modification with ubiquitin, including ~600 E3 ligases. We report here that E6AP/UBE3A is distinguished from other E3 ligases by having a 12 nM binding site at the proteasome contributed by substrate receptor hRpn10/PSMD4/S5a. Intrinsically disordered by itself, and previously uncharacterized, the E6AP-binding domain in hRpn10 locks into a well-defined helical structure to form an intermolecular 4-helix bundle with the E6AP AZUL, which is unique to this E3. We thus name the hRpn10 AZUL-binding domain RAZUL. We further find in human cells that loss of RAZUL by CRISPR-based gene editing leads to loss of E6AP at proteasomes. Moreover, proteasome-associated ubiquitin is reduced following E6AP knockdown or displacement from proteasomes, suggesting that E6AP ubiquitinates substrates at or for the proteasome. Altogether, our findings indicate E6AP to be a privileged E3 for the proteasome, with a dedicated, high affinity binding site contributed by hRpn10.

---

[1] Protein Processing Section, Structural Biophysics Laboratory, Center for Cancer Research, National Cancer Institute, Frederick, MD 21702, USA. [2] Genome Modification Core, Frederick National Laboratory for Cancer Research, Frederick, MD 21702, USA. [3] Protein Characterization Laboratory, Frederick National Laboratory for Cancer Research, Frederick, MD 21702, USA. [4] Biophysics Resource, Structural Biophysics Laboratory, Center for Cancer Research, National Cancer Institute, Frederick, MD 21702, USA. [5] Laboratory of Cancer Immunometabolism, Center for Cancer Research, National Cancer Institute, Frederick, MD 21702, USA. [6] These authors contributed equally: Gwen R. Buel, Xiang Chen. ✉email: xiang.chen@nih.gov; kylie.walters@nih.gov

The 26S proteasome is a 2.5 MDa complex responsible for regulated protein degradation[1,2], with substrates typically ubiquitinated by a hierarchical enzymatic cascade[3]. An E1 activating enzyme charges ubiquitin to become a protein modifier and transfers it to an E2 conjugating enzyme which, in concert with an E3 ligating enzyme, subsequently attaches ubiquitin to a substrate. Approximately 600 E3s exist in humans that can either accept the charged ubiquitin for direct transfer to a substrate or serve as a scaffold for ubiquitin transfer from the E2 to a substrate[4–6]. Following ubiquitination, receptor sites in the proteasome contributed by Rpn1, Rpn10, and Rpn13 recognize ubiquitin directly or the ubiquitin fold of shuttle factor ubiquitin-like domains[7–14]; shuttle factors bind ubiquitinated substrates by one or more ubiquitin-associated domain[15–17]. At the proteasome, ubiquitin chains are hydrolyzed by deubiquitinating enzymes (DUBs) Rpn11[18], UCHL5/Uch37[19], and Usp14[20–22], as the marked substrate is translocated through an ATPase ring for entry into the hollow interior of the proteolytic core particle (CP)[2,23–25]. The integrity of the ubiquitin-proteasome pathway is essential for cellular homeostasis with dysfunction linked to disease, including cancer and neurodegeneration. Inhibitors of the CP are used to treat hematologic cancers[26–28] and additional proteasome subunits are being pursued as synergistic targets, including hRpn13[29–34].

Hijacking of ubiquitin E3 ligase E6AP/UBE3A by high risk human papilloma virus E6 oncoprotein contributes to cervical cancer by inducing ubiquitination and in turn degradation of tumor suppressor p53[35–37]. Moreover, loss-of-function mutations in E6AP associate with Angelman syndrome[38–40] and elevated gene dosage with autism spectrum disorders[41]. How aberrant E6AP mechanistically contributes to these neurological diseases is an active area of investigation; however, E6AP was recently found to ubiquitinate calcium- and voltage-dependent potassium channels, the dysfunction and hyperexcitability of which is associated with Angelman syndrome[42]. E6AP distributes in an isoform-dependent manner between the nucleus and cytosol of neurons[43,44] and contains an N-terminal Zn-binding AZUL (amino-terminal zinc-binding domain of ubiquitin E3a ligase) domain[45] that binds to Rpn10 in the proteasome[46], and is required for E6AP nuclear localization[44]. An Angelman syndrome-associated missense mutation in the E6AP HECT domain interferes with E6AP nuclear localization[44], although the connection between this mutation and the requirement for the AZUL domain is not known. E6AP also stimulates Wnt/β-catenin signaling, a function that requires its ubiquitin ligase activity and interaction with the proteasome, and is disrupted by an autism-linked E6AP mutation[46,47].

Although multiple E3 ligases have been reported to associate with the proteasome[22,48,49], no E3 ligase-proteasome complex structure is available, nor has any domain been identified for recruiting an E3 ligase to the proteasome. Here, we identify a region at the C-terminal end of hRpn10 that forms a binding site in the proteasome for E6AP. By using biophysical techniques including NMR spectroscopy, we find this region to be disordered when unbound, but upon binding to E6AP, to fold into an independent structural domain characterized by two helices that pack against the E6AP AZUL to form a 4-helix bundle. To test the significance of the E6AP-binding domain, we used gene editing to generate cell lines in which it is deleted and find that this hRpn10 domain contributes E6AP to the proteasome. We unexpectedly find that hRpn10 levels are coupled to E6AP cellular protein levels. Altogether, our data suggest a dual regulatory role for hRpn10 in E6AP function and that through hRpn10, E6AP is a privileged ubiquitin E3 ligase for the proteasome.

## Results

**Human Rpn10 contains a C-terminal domain that binds E6AP.** Rpn10 has an N-terminal von Willebrand factor type A (VWA) domain that assembles into the proteasome and a UIM region for binding ubiquitinated proteins; these domains complete the protein in fungi (Supplementary Fig. 1a). In higher eukaryotes, Rpn10 contains an additional conserved ~70 amino acids at the C-terminus (Fig. 1a and Supplementary Fig. 1a). We recorded a 2D NMR experiment on $^{15}$N-hRpn10 spanning 196–377, which encompasses the UIM region and uncharacterized C-terminal end. The resulting spectrum indicated that the UIM region, readily identified by our previous assignment of these amino acids[50], is unperturbed by the additional C-terminal sequence (Supplementary Fig. 1b). Ubiquitin addition to $^{15}$N-hRpn10$^{196–377}$ demonstrated expected shifting of the UIMs[51,52] but no effect for the unassigned signals (Supplementary Fig. 1c). Thus, the additional sequence does not interact with the UIM region, nor does it bind ubiquitin.

To identify proteins that interact directly or in complex with the C-terminal hRpn10 domain, we subcloned the region spanning 305–377 in frame with GST, expressed and purified the fusion protein from *Escherichia coli*, and bound purified GST-hRpn10$^{305–377}$ to glutathione sepharose resin for incubation with lysates from 293T (human embryonic kidney epithelial) or HCT116 (colorectal carcinoma) cells. After washing, resin-bound proteins were separated by SDS-PAGE, eluted from the gel, digested with trypsin, and analyzed by mass spectrometry; parallel experiments were done with GST protein as a control. The only hit identified for either 293T or HCT116 lysate was E6AP (Supplementary Fig. 2a).

The human *UBE3A* gene encodes three isoforms generated by differential splicing[53] that vary at the extreme N-terminus (Supplementary Fig. 2b). To test whether E6AP binds to the hRpn10 C-terminal region directly, we incubated full-length E6AP (Isoform II) with Ni-NTA resin pre-bound to His-hRpn10$^{full-length}$, His-hRpn10$^{196–377}$, or His-hRpn10$^{196–306}$. hRpn10$^{full-length}$ and hRpn10$^{196–377}$ bound E6AP, whereas hRpn10$^{196–306}$ did not (Fig. 1b). We next added unlabeled AZUL, which is present in all three E6AP isoforms (Supplementary Fig. 2b), to $^{15}$N-hRpn10$^{305–377}$ and monitored the effect by 2D NMR to find hRpn10 signals shifted (Fig. 1c), indicating binding. We assigned both the free and AZUL-bound state, as described in "Methods", and quantified the changes to find D327–M356 perturbed (Supplementary Fig. 2c). Analogous experiments with unlabeled hRpn10$^{305–377}$ and $^{15}$N-E6AP$^{AZUL}$, aided by previous assignments[45], indicated residues in both AZUL helices to be significantly shifted by hRpn10 addition (Supplementary Fig. 2d, e). Altogether, these experiments indicate direct binding between E6AP$^{AZUL}$ and hRpn10$^{305–377}$, consistent with recent publications implicating AZUL and hRpn10 to E6AP interaction with the proteasome[44,46].

To assess the strength of hRpn10$^{305–377}$:AZUL interaction, we used isothermal titration calorimetry (ITC) with the AZUL added incrementally to hRpn10$^{305–377}$; a $K_d$ value of 11.6 + 3.3 nM was measured (Fig. 1d and Supplementary Fig. 2f), indicating similar strength to hRpn13 interaction with the proteasome[54,55]. Surface plasmon resonance (SPR) similarly revealed a $K_d$ value of 8.1 + 1.4 nM for GST-hRpn10$^{305–377}$ binding to AZUL (Fig. 1d and Supplementary Fig. 2g).

To test whether the hRpn10 C-terminal region is required for interaction with endogenous E6AP in cells, lysates from HCT116 cells expressing either myc-hRpn10$^{full-length}$ or myc-hRpn10$^{1–306}$ were subjected to immunoprecipitation with anti-myc nanobody-coupled agarose. Co-immunoprecipitation of E6AP was observed with full-length but not truncated

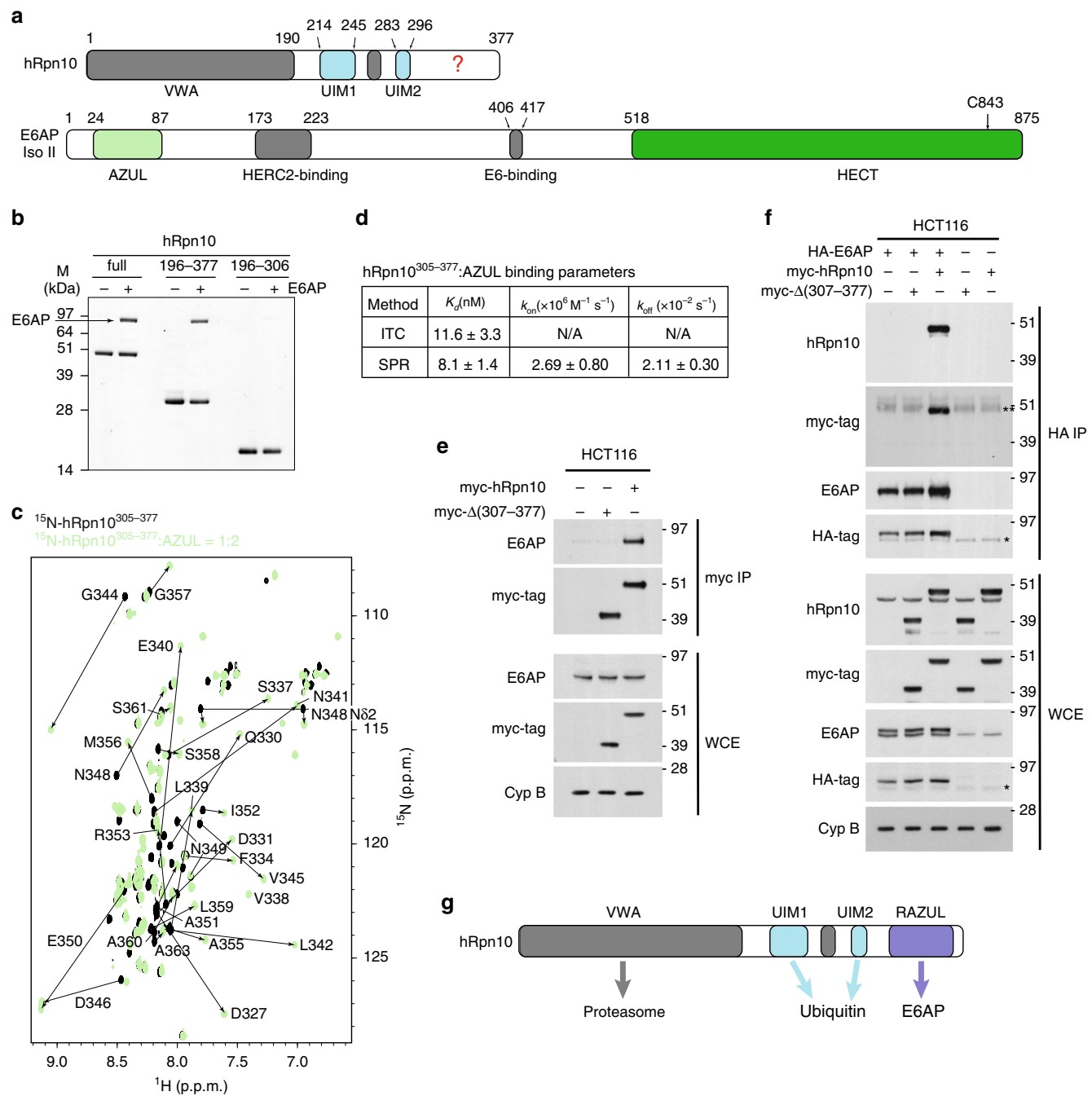

**Fig. 1 A C-terminal domain in hRpn10 binds E6AP AZUL. a** Positions of known functional domains within hRpn10 (top) and E6AP isoform II (bottom). Question mark "?" indicates hRpn10 uncharacterized region and the E6AP catalytic cysteine C843 is indicated. **b** Pull-down assay of His-tagged hRpn10[full-length] (full), hRpn10[196–377] or hRpn10[196–306] without (−) or with (+) incubation of E6AP. **c** $^{1}$H, $^{15}$N HSQC spectra of 0.2 mM $^{15}$N-hRpn10[305–377] (black) and with twofold molar excess unlabeled AZUL (green). Shifted signals are labeled. **d** Table summarizing $K_d$, $k_{on}$, and $k_{off}$ average values with standard deviations for the hRpn10[305–377]: AZUL interaction measured by ITC and/or SPR. N/A, not applicable. **e** HCT116 lysates expressing empty vector, myc-hRpn10 full length, or myc-Rpn10 with RAZUL deleted (Δ307–377) were subjected to myc-immunoprecipitation with anti-myc-tag nanobody-coupled agarose. Whole cell extracts (WCE) and myc-immunoprecipitates were immunoprobed with the indicated antibodies. Cyclophilin B (Cyp B) is used as a loading control in **e** and **f**. **f** Lysates from HCT116 cells expressing HA-E6AP and the myc-hRpn10 constructs of **e** were subjected to HA IP followed by immunoblotting with the indicated antibodies. An asterisk "*" indicates non-specific interaction; double asterisk "**" indicates heavy chain antibody. **e–f** All antibodies used for immunoprobing are indicated to the left of the images. Note that the hRpn10 and E6AP antibodies recognize both endogenous and exogenously expressed protein, causing these panels to show both tagged and endogenous protein. **g** Schematic representation highlighting interaction domains of hRpn10 including newly identified RAZUL. Source data are provided as a Source Data file.

hRpn10 (Fig. 1e). We further co-expressed HA-E6AP with either hRpn10 construct and immunoprecipitated E6AP with anti-HA antibodies to find co-immunoprecipitation of full-length, but not truncated, hRpn10 (Fig. 1f).

Altogether, these data indicate that the E6AP AZUL is a strong interaction partner of hRpn10[305–377] and we henceforth refer to this domain in Rpn10 as RAZUL (Rpn10 AZUL-binding domain, Fig. 1g).

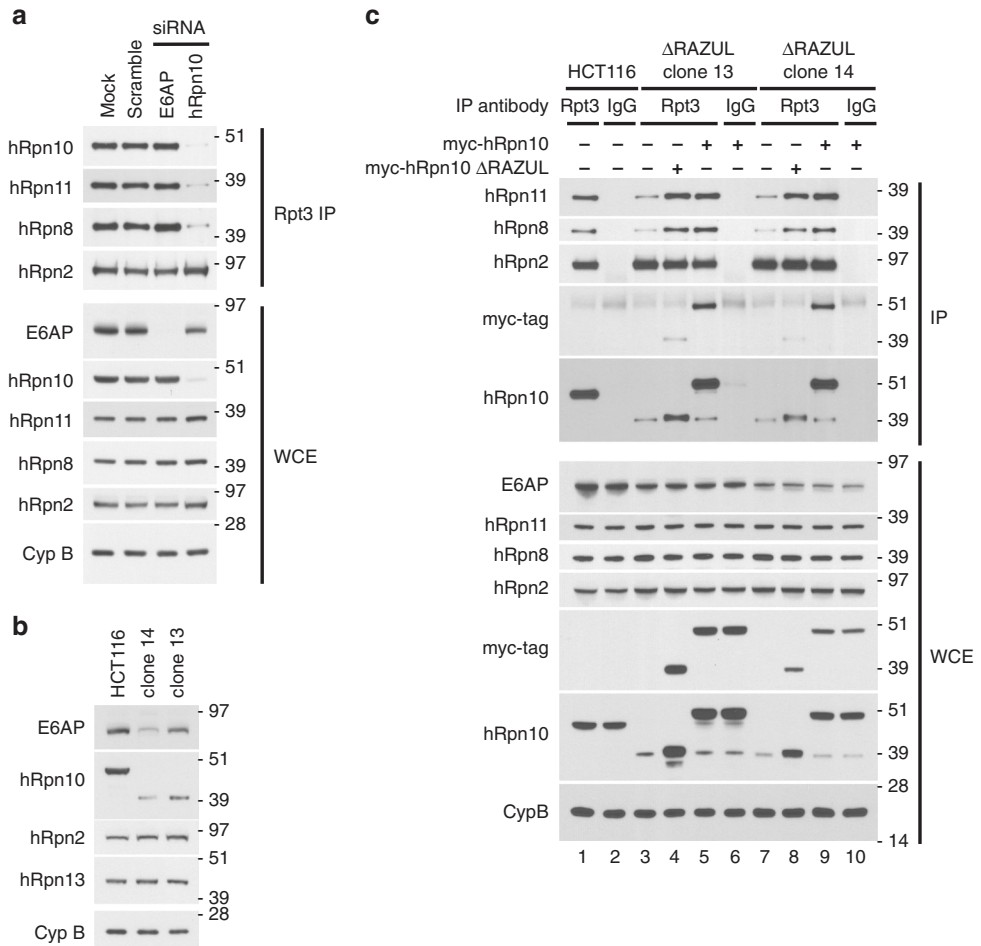

**Fig. 2 Levels of proteasome lid-base association correlate with hRpn10 levels. a** Lysates from HCT116 cells transfected with siRNAs to E6AP or hRpn10, as well as mock and scrambled control samples, were immunoprecipitated with Rpt3 antibodies. WCE and immunoprecipitates were immunoprobed as indicated. Cyclophilin B (Cyp B) is used as a loading control in **a–c**. **b** Immunoblots of parental HCT116 cells and two clonal cell lines (clone 14 and clone 13) generated by CRISPR-mediated truncation of hRpn10. **c** ΔRAZUL cells shown in **b** were transfected with full length or ΔRAZUL hRpn10 constructs and subjected to hRpt3 immunoprecipitation. Lysates or immunoprecipitates were probed with antibodies as indicated to the left of each panel. Source data are provided as a Source Data file.

**E6AP protein levels depend on hRpn10.** To test the significance of the RAZUL:AZUL interaction, we considered generating a full hRpn10 knockout cell line. However, we found that hRpn10 knockdown by siRNA resulted in loss of proteasome components hRpn8 and hRpn11 associating with proteasome ATPase Rpt3 (Fig. 2a), consistent with an early report that Rpn10 is necessary for base-lid interactions in the yeast proteasome[56]. To avoid such proteasome defects, we utilized CRISPR/Cas9 to generate a cell line in which hRpn10 lacks RAZUL but retains intact VWA and UIMs (see "Methods"). Deletion of RAZUL was assayed by immunoblotting, as demonstrated for clones 13 and 14 (Fig. 2b); we henceforth refer to these cell lines as *ΔRAZUL*.

Lower levels of truncated hRpn10 were consistently observed in *ΔRAZUL* cells compared with the full-length protein in *WT* cells, with clone 14 producing less protein than clone 13 (Fig. 2b, second panel). Corresponding with the lower levels, association of truncated hRpn10 with proteasome immunoprecipitated by anti-hRpt3 antibodies was reduced in the *ΔRAZUL* cell lines compared with HCT116 cells (Fig. 2c, lanes 1, 3, and 7). Moreover, co-immunoprecipation with Rpt3 of lid components hRpn8 and hRpn11, and not of base component Rpn2, was similarly reduced (Fig. 2c, lanes 1, 3, and 7). Expression of either myc-hRpn10[full-length] or myc-hRpn10[ΔRAZUL] rescued the association

of these lid components in both *ΔRAZUL* cell lines (Fig. 2c, lanes 4, 5, 8, and 9). This finding establishes that RAZUL is not necessary for lid association with the proteasome base, and that levels of hRpn10 correlate with levels of proteasome lid-base association.

In addition to hRpn10 being important for proteasome base-lid assembly, we unexpectedly found that E6AP levels correlated with hRpn10 ΔRAZUL levels (Fig. 2b, top panel). To test whether the reduced protein levels are due to protein degradation, *WT* and *ΔRAZUL* cells were treated with 10 μM MG132 for 4 h to inhibit the proteasome. As expected, MG132 treatment caused ubiquitinated proteins to accumulate in both cell lysates (Supplementary Fig. 3, third panel). No increase was observed however for either hRpn10 ΔRAZUL or E6AP, nor were higher molecular weight bands apparent (Supplementary Fig. 3). We assayed E6AP levels in cells with siRNA knockdown of hRpn10 to similarly find direct correlation (Fig. 3a). Loss of E6AP by siRNA treatment however had no effect on hRpn10 levels (Fig. 3b).

We attempted to rescue E6AP levels in *ΔRAZUL* cells (clone 14) by expressing myc-hRpn10[full-length] or myc-hRpn10[ΔRAZUL]; however, no increase in endogenous E6AP protein was observed for either condition (Fig. 3c). We further interrogated this effect in *WT* or *ΔRAZUL* cells by expressing E6AP and hRpn10 constructs

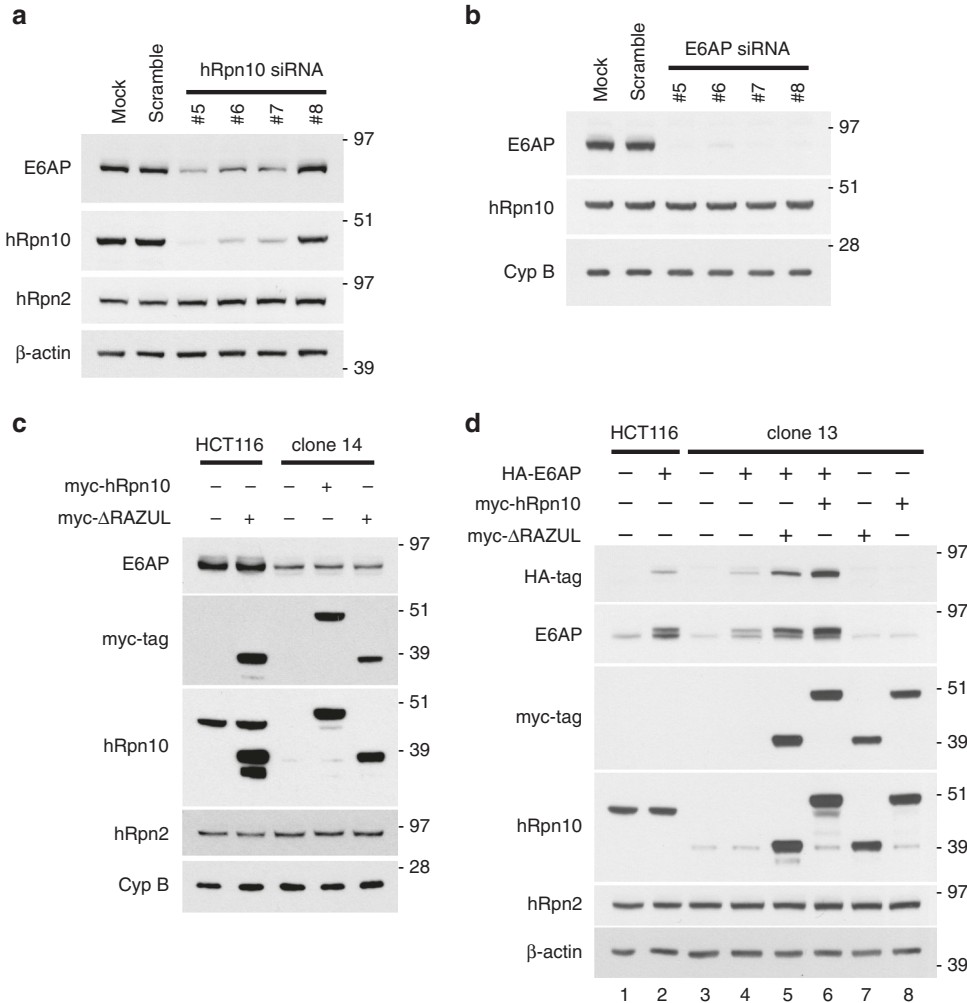

**Fig. 3 E6AP levels depend on hRpn10.** hRpn10 (**a**) or E6AP (**b**) was knocked down in HCT116 cells by four different siRNAs and the cell lysates immunoprobed as indicated. Mock and scrambled control samples are included. β-actin is used as a loading control in **a** and **d**. **c** Lysates from HCT116 or clone 14 cells expressing myc-hRpn10 constructs were immunoprobed as indicated. **d** Lysates from HCT116 or clone 13 cells expressing HA-E6AP and/or myc-hRpn10 constructs were immunoprobed as indicated. **a–d** Antibodies used for immunoprobing are indicated to the left of each panel. Source data are provided as a Source Data file.

either independently or in combination. Exogenous expression of E6AP was reduced in *ΔRAZUL* (clone 13) compared with *WT* cells (Fig. 3d, lane 2 vs. lane 4). Exogenous E6AP levels were boosted however in *ΔRAZUL* cells (clone 13) when either truncated (lane 5) or full-length (lane 6) hRpn10 was co-transfected with E6AP (Fig. 3d); thus the effect is not dependent on the RAZUL: AZUL interaction, however full-length hRpn10 had a stronger effect on E6AP levels than truncated hRpn10. Altogether, these data link hRpn10 to E6AP production with possible spatial or temporal regulation that allows rescue of E6AP when the two proteins are co-expressed.

**E6AP AZUL binds to hRpn10 RAZUL at the proteasome**. We used *ΔRAZUL* cells to test whether the hRpn10 RAZUL recruits E6AP to the proteasome, boosting hRpn10 levels by transient transfection. Proteasomes from lysates of *WT* and *ΔRAZUL* cells (clone 13) expressing myc-tagged hRpn10$^{full-length}$ or ΔRAZUL protein were immunoprecipitated with Rpt3 antibodies and immunoprobed for hRpn10, E6AP, or proteasome component hRpn2 (as a control). E6AP co-immunoprecipitated with proteasomes from *WT* (Fig. 4a, lane 2 and 3), but not *ΔRAZUL* (Fig. 4a, lane 5) cells. Expression of full-length (lane 7) but not RAZUL-truncated hRpn10 (lane 6) resulted in

observable E6AP co-immunoprecipitation with proteasomes isolated from *ΔRAZUL* cells (Fig. 4a); we attribute the lower amounts of E6AP co-immunoprecipitated with proteasomes of hRpn10$^{full-length}$-expressing *ΔRAZUL* cells to the reduced abundance of endogenous E6AP in this cell line, as described above (Fig. 2b). This experiment indicates that the hRpn10 RAZUL recruits E6AP to the proteasome.

We further tested whether RAZUL could compete with the proteasome for E6AP binding. Proteasomes immunoprecipitated from HCT116 cells overexpressing myc-hRpn10 RAZUL were immunoprobed for E6AP and compared with empty vector transfected cells and an IgG control. Expression of RAZUL caused E6AP to be lost from the proteasome (Fig. 4b).

To test the impact of E6AP at the proteasome, Rpt3 immunoprecipitates from lysates of cells transfected with a scrambled control or siRNA against E6AP were immunoprobed for ubiquitin. While no major change in total ubiquitin levels was observed at the level of whole cell extract (WCE), a reduction in bulk ubiquitin was apparent at the proteasome following E6AP loss, particularly for higher molecular weight species (Fig. 4c). To test whether displacing E6AP from the proteasome has a similar effect on ubiquitin levels, we expressed RAZUL to compete E6AP away from endogenous hRpn10. Ubiquitin associating with the

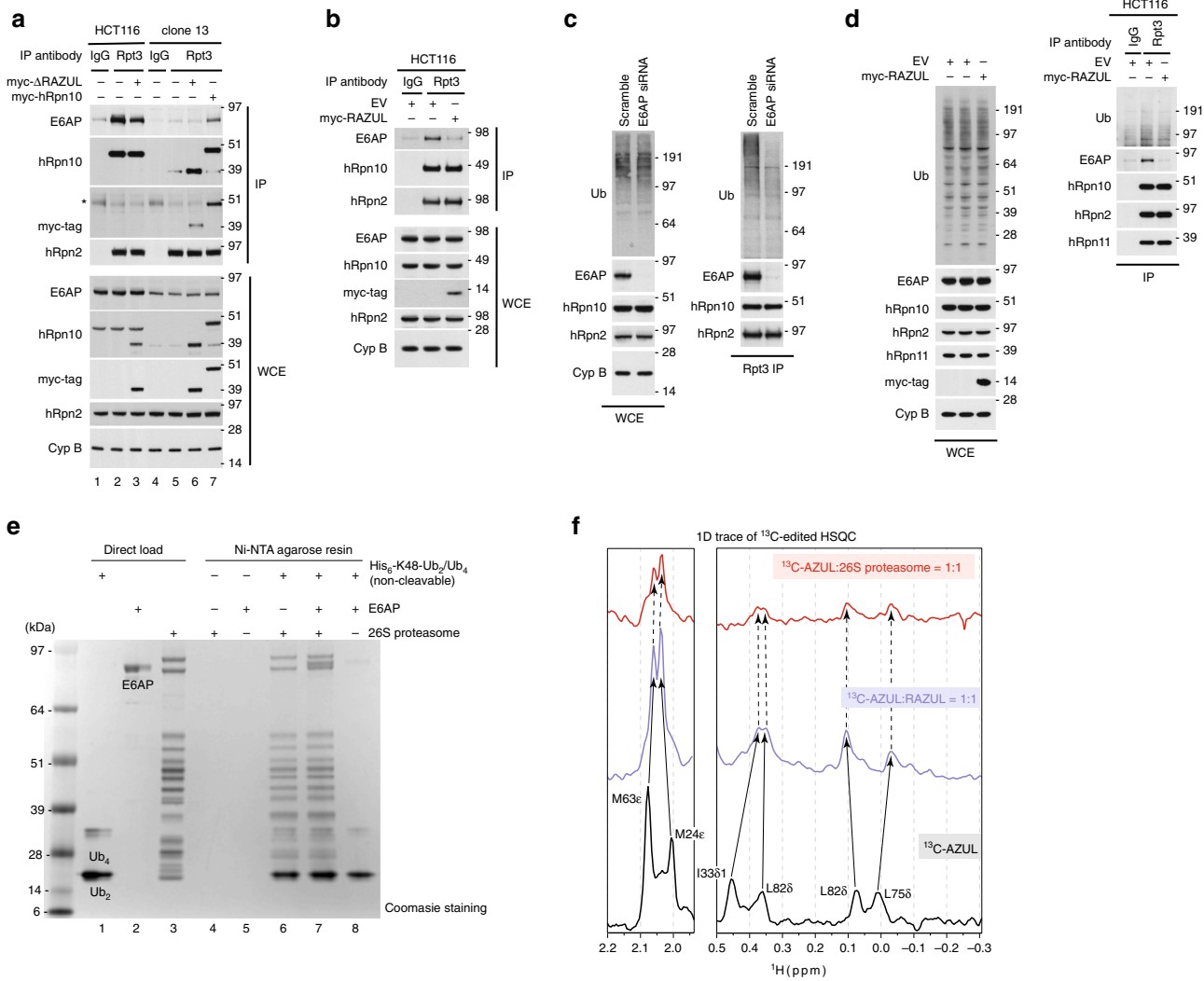

**Fig. 4 hRpn10 RAZUL contributes E6AP to the proteasome. a** Immunoblots of Rpt3 immunoprecipitates or WCE from HCT116 or clone 13 lysates expressing myc-hRpn10 constructs. An asterisk "*" indicates heavy chain antibody. Cyclophilin B (Cyp B) is used as a loading control for WCE samples in **a–c** and hRpn2 as a positive control for the immunoprecipitation. IgG controls are included. **b, d** Immunoblots of Rpt3 or IgG (control) immunoprecipitates or WCE of lysates from HCT116 cells transfected with empty vector (as a control) or myc-hRpn10 RAZUL. **c** Immunoblots of Rpt3 immunoprecipitates or WCE from lysates of HCT116 cells transfected with a scrambled control or siRNA against E6AP. **a–d** Antibodies used for immunoprobing are indicated to the left of each panel. **e** Pull-down assay for a commercially available mixture of His$_6$-tagged, non-cleavable K48-linked Ub$_2$/Ub$_4$ with incubation of human 26S proteasome (lane 6), 26S proteasome with equimolar E6AP (lane 7), or just E6AP (lane 8). E6AP or 26S proteasome was added to Ni-NTA agarose resin as negative controls (lanes 4 and 5). K48-linked Ub$_2$/Ub$_4$, E6AP, and 26S proteasome were loaded directly in lanes 1–3, as indicated. **f** Selected regions from 1D $^{13}$C-edited, $^1$H NMR experiments acquired at 850 MHz and 25 °C for free $^{13}$C-AZUL (black) or mixtures with equimolar unlabeled RAZUL (blue) or 26S proteasome (red). The concentration of each sample was 0.3 μM and 200,000 scans were recorded for each experiment. Source data are provided as a Source Data file.

proteasome was decreased in the RAZUL-expressing condition (Fig. 4d), further correlating E6AP interaction with hRpn10 to ubiquitin levels at the proteasome.

We next tested whether E6AP impacts the affinity of the 26S proteasome for ubiquitin chains by using an in vitro assay. 26S proteasome was added alone or with equimolar E6AP to Ni-NTA resin pre-bound to a commercially available mixture of non-hydrolyzable K48-linked Ub$_2$/Ub$_4$. The amount of proteasome retained on the resin following extensive washing was unaffected by the presence of E6AP (Fig. 4e, lane 7 compared with lane 6). Moreover, E6AP was retained on the ubiquitin-bound resin when proteasome was present (Fig. 4e, lane 7) and barely observable without proteasome (Fig. 4e, lane 8). Thus, the loss of ubiquitinated protein at the proteasome following E6AP knockdown (Fig. 4c) is not caused by a

reduction of E6AP-bound proteasome affinity for ubiquitin chains.

hRpn10 is assembled into the proteasome RP at equimolar stoichiometry through its N-terminal VWA domain, which is well-defined in cryoelectron microscopy structures of the proteasome; however, the remaining portion of hRpn10, including the ubiquitin-binding UIM portion, is missing from reported cryoelectron microscopy densities[57–63] due to flexibility[51]. To test directly whether the RAZUL:AZUL interaction observed for the isolated domain complex is maintained in the intact proteasome, we used NMR, which is ideal for flexible systems[64,65]. We added equimolar $^{13}$C-AZUL to RAZUL or 26S proteasome (Supplementary Fig. 4a) and acquired a 1D $^{13}$C-edited, $^1$H NMR experiment for comparison with free $^{13}$C-AZUL. Binding to RAZUL induced shifting (Supplementary Fig. 4b, middle vs.

**Table 1 Structural statistics for the RAZUL:AZUL complex.**

|  | RAZUL | AZUL |
|---|---|---|
| *NMR distance and dihedral constraints* |  |  |
| NOE-derived distance constraints |  |  |
| Intramolecular | 705 | 1084 |
| Intra-residue | 386 | 412 |
| Inter-residue | 319 | 672 |
| Sequential ($|i - j| = 1$) | 192 | 257 |
| Nonsequential ($|i - j| > 1$) | 127 | 415 |
| Intermolecular | 217 |  |
| Hydrogen bonds | 15 | 22 |
| Total dihedral angle constrains | 66 | 110 |
| ϕ | 33 | 55 |
| ψ | 33 | 55 |
| *Structure statistics* |  |  |
| Violations (mean and s.d.) |  |  |
| Distance constraints (Å) | 0 |  |
| Dihedral angle constraints (º) | 0 |  |
| Max. dihedral angle violation (º) | <5 |  |
| Max. distance constraint violation (Å) | <0.3 |  |
| Deviations from idealized geometry |  |  |
| Bond lengths (Å) | 0.002 ± 0.000 |  |
| Bond angles (º) | 0.416 ± 0.008 |  |
| Impropers (º) | 0.283 ± 0.014 |  |
| Average pairwise r.m.s.d.* (Å) |  |  |
| Backbone atoms | 0.59 ± 0.11 |  |
| Heavy atoms | 1.24 ± 0.17 |  |

*Pairwise r.m.s.d. for the 15 lowest energy structures for V328-A360 from RAZUL and K25-L82 from AZUL.

bottom panel), as exemplified by methyl groups of M24 and M63 (Fig. 4f, left panel) and I33, L75, and L82 (Fig. 4f, right panel). The spectrum acquired with proteasome added closely mimicked that with only RAZUL added (Fig. 4f and Supplementary Fig. 4b), indicating that AZUL binding to RAZUL occurs identically at the proteasome. Compared with the spectrum recorded with only RAZUL added, the AZUL signals were broadened by proteasome addition, but not to the extent expected for a 2.5 MDa complex[66,67]. Rather, the appearance of proteasome-bound AZUL signals provides strong support for the AZUL:RAZUL subcomplex being tethered to but not docked against the rest of the 26S proteasome.

**E6AP induces helicity in RAZUL**. We used NMR to solve the structure of the RAZUL:AZUL complex, as described previously[64,67] and in "Methods", with the data summarized in Table 1. We recorded [13]C-half-filtered NOESY experiments on samples of the complex with one protein [13]C labeled and the other unlabeled to measure unambiguous intermolecular interactions (Supplementary Fig. 5). Altogether, 217 interactions between AZUL and RAZUL were identified (Table 1). The 15 lowest energy structures converged with a root-mean-square deviation (r.m.s.d.) of 0.59 Å (Fig. 5a).

Unbound AZUL comprises two helices (H1 and H2) and a Zn finger[45] and this architecture is maintained in the RAZUL-bound state, with a backbone r.m.s.d. between the free and complexed structures of 1.3 Å (Supplementary Fig. 6a). RAZUL binds the AZUL helices from the opposite direction compared with the Zn finger (Fig. 5b). In this complex, two α-helices are formed in RAZUL that span P332–N341 (α1) and E350–S361 (α2), with an angle of 158.5º between the two helical axes. Directly N-terminal to RAZUL α1 is a single turn of a $3_{10}$-helix that spans V328-Q330 (Fig. 5b). We submitted the atomic coordinates of the RAZUL: AZUL complex to the Dali server[68] to find no similarities with

either AZUL or RAZUL, indicating that RAZUL is unique and not previously described.

Comparisons of our NMR data acquired on free and AZUL-bound RAZUL indicate that RAZUL acquires helicity upon binding to AZUL. Carbonyl and Cα values when compared with random coil taking into account amino acid type yields a chemical shift index (CSI) that informs on secondary structure; these values shift to reflect greater helicity for RAZUL when bound to AZUL (Fig. 5c and Supplementary Fig. 6b). Moreover, intramolecular interactions characteristic of helicity were observed following AZUL addition, but not for free RAZUL (Supplementary Fig. 6c, d). Overall, our NMR data indicate that RAZUL switches from a poorly ordered state to a well-defined helical state following AZUL binding. We interrogated this finding further by circular dichroism (CD) spectroscopy, as done previously for Rap80[69,70]. CD measurements indicated 7% and 46% helicity, respectively, for unbound RAZUL (blue) and AZUL (green), and a theoretical spectrum (gray dashed line) for the mixture with markedly less spectral features of helicity compared with the recorded experimental spectrum (orange, Fig. 5d). Overall, 35% helicity is indicated from the experimental CD data recorded on the complex, consistent with the 36% helicity determined by NMR (Fig. 5b). The AZUL secondary structure is unaltered by binding to RAZUL (Supplementary Fig. 6a), leading us to conclude that the observed difference between the theoretical and experimental CD spectra reflects increased helicity for RAZUL, consistent with the NMR data (for example, Fig. 5c).

**Structure of the RAZUL:AZUL complex**. At the molecular interface, a 4-helix bundle is formed by two pairs of helices from AZUL and RAZUL stacking against each other (Fig. 5b). RAZUL α1 is centered between the two AZUL helices by hydrophobic interactions involving F334, L335, V338, and L339 as well as L342 and V345 from the RAZUL α1/ α2 loop (Fig. 6a, b). These residues interact with A29, I33, and Y37 from AZUL H1 and A69 and L73 from AZUL H2 (Fig. 6a, b). From the $3_{10}$-helix, V328 and M329 form hydrophobic interactions with AZUL A29, L73, and Y76 (Fig. 6c), capping the hydrophobic contact surface formed by RAZUL α1. RAZUL α2 is more peripheral compared with α1, with A351, I352, A355, M356, and L359 interacting with A67, L70 and L73 from AZUL H2 (Fig. 6d).

The RAZUL N-terminal end (E322–D327) is rich in acidic residues (Supplementary Fig. 1a) and proximal to the positively charged AZUL N-terminal end, which includes K25 and R26 (Fig. 6e). These AZUL residues contribute three hydrogen bonds to the complex, engaging RAZUL E323, D324, and Y326 (Fig. 6c). Y326, which is phosphorylated in Jurkat cells[71], forms a hydrogen bond with the AZUL R26 sidechain in 80% of calculated structures, as well as hydrophobic contacts with AZUL K25 and R26 (Fig. 6c). We tested whether adding a bulky phosphate group at this location could be deleterious for AZUL binding by synthesizing RAZUL peptides that span E322–D366 without and with Y326 phosphorylated and measuring affinity by ITC. This shorter wild-type peptide bound with an affinity within error of hRpn10[305–377] (Fig. 6f and Supplementary Fig. 7a), as expected from the structure (Fig. 5a, b). Y326 phosphorylation of hRpn10[322–366] reduced affinity for AZUL tenfold (Fig. 6f and Supplementary Fig. 7b). We next tested the importance of the hydrogen bond contributed by Y326 by replacing this amino acid with phenylalanine. E6AP co-immunoprecipitation with the proteasome is reduced in ΔRAZUL cells expressing hRpn10 Y326F compared with those expressing the wild-type protein (Fig. 6g, lane 8 vs. 7). We also found that losing the unstructured C-terminal region (365–377) from hRpn10 had no effect on E6AP co-immunoprecipitation with proteasomes by expressing

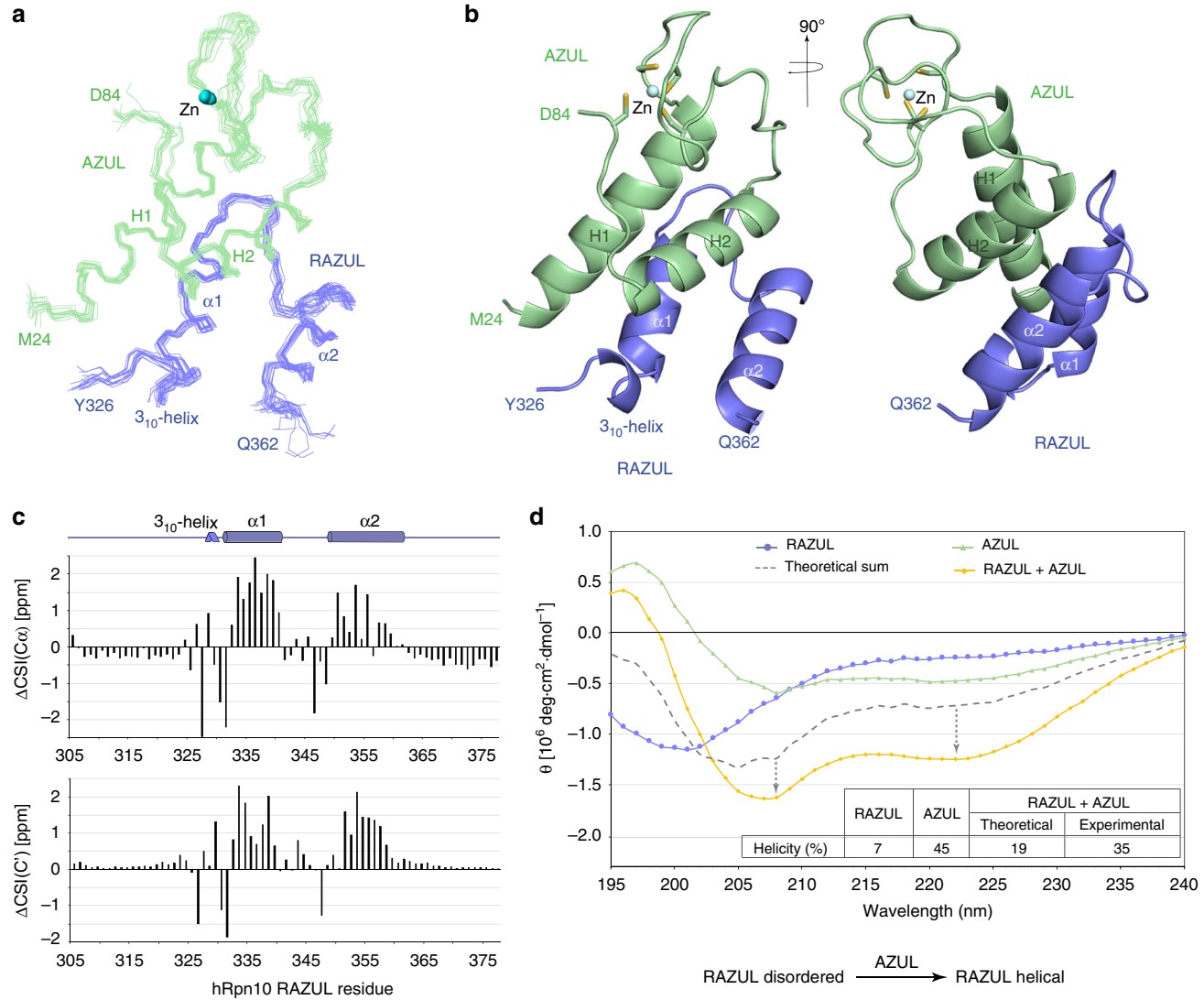

**Fig. 5 AZUL induces helicity in hRpn10 RAZUL. a** Backbone trace labeling terminal residues of 15 lowest energy RAZUL (blue):AZUL (green) structures superimposing secondary structures with Zn displayed (blue sphere). Secondary structures and Zn are labeled in **a**, **b**. **b** Representative ribbon diagram of the RAZUL:AZUL structure with sidechain atoms of Zn-coordinating cysteines displayed as sticks with sulfur yellow. **c** AZUL-induced CSI change (ΔCSI) for RAZUL Cα (top) and C′ (bottom). $\Delta CSI(C\alpha) = CSI(C\alpha_{bound}) - CSI(C\alpha_{free})$; $\Delta CSI(C') = CSI(C'_{bound}) - CSI(C'_{free})$. **d** CD spectra of RAZUL (blue), AZUL (green), and the mixture (orange). The theoretical sum spectrum based on free values (gray) is displayed, with a table listing predicted helicity and schematic depicting the effect of AZUL on RAZUL structure. Source data are provided as a Source Data file.

hRpn10 (1–364) in ΔRAZUL cells (Fig. 6g, lane 9). This finding is consistent with the ITC data indicating equivalent AZUL affinity for hRpn10[322–366] (Fig. 6f).

Altogether, these data indicate that the hRpn10 RAZUL forms an independent structural domain that contributes E6AP to the proteasome.

## Discussion

Approximately 600 ubiquitin E3 ligases exist in humans. We report here that proteasome substrate receptor hRpn10 evolved a 12 nM affinity binding domain for recruiting E6AP to the proteasome through the N-terminal AZUL domain, which is a unique feature of E6AP (Fig. 7). This result establishes E6AP as a privileged E3 ligase for the proteasome with the possibility of direct coupling of its ligase activity to proteasome activity. This finding provides new foundational knowledge that impacts future studies aimed at addressing the role of E6AP in cervical cancer, Angelman syndrome, and autism. Our results also redefine the current models of the 26S proteasome to include a dedicated

binding domain in hRpn10 that has until now been uncharacterized despite the discovery of Rpn10 as a proteasome substrate receptor over 2.5 decades ago[72]. This domain may have remained elusive in earlier studies in part due to its disordered state when unbound and its absence in fungi.

The induced folding of RAZUL upon binding to E6AP is similar to the coupled folding reported in previous studies[73,74], such as the N-terminal transactivation (TAD) domain of p53, which exchanges between a disordered and partially helical conformation when unbound[75–77] and forms a stable amphipathic α-helix when complexed with the E3 ligase Mdm2[78]. Increasing intrinsic helicity of the p53 TAD domain yields stronger binding to Mdm2 in vitro and in cells[79], suggesting that RAZUL could potentially be engineered to increase E6AP occupancy at the proteasome in future work or alternatively for E6AP targeting.

In contrast to hRpn10, the other two proteasome receptors, Rpn1 and Rpn13, contribute binding sites for DUBs[11,22,80–82]. Proximity between substrate receptors and DUBs is no doubt

**Fig. 6 RAZUL:AZUL forms an intermolecular 4-helix bundle.** Regions displaying AZUL contacts with RAZUL α1 (**a**, **b**), the 3₁₀-helix (**c**), and α2 (**d**) showing selected interacting sidechain atoms with oxygen, nitrogen, and sulfur colored red, blue, and yellow, respectively. The structure in **c** is displayed as a ribbon diagram to better view the $3_{10}$-helix with hydrogen bonds included as red dashed lines. The structures in **a**, **b**, and **d** depict helices as cylinders for simplicity. **e** Electrostatic surface diagram for AZUL to highlight acidic (red) and basic (blue) regions. The bound RAZUL is displayed with cylindrical helices and sticks for N-terminal acidic residues with oxygen colored red. **f** ITC analyses for AZUL-binding hRpn10$^{322-366}$ without or with Y326 phosphorylated. **g** Immunoprobing of Rpt3 immunoprecipitates or WCE from HCT116 or clone 13 lysates expressing indicated myc-hRpn10 constructs, including with RAZUL (ΔRAZUL) or the C-terminal 13 residues deleted (1–364) or with hRpn10 Y326 substituted with phenylalanine (Y326F). Cyclophilin B (Cyp B) is used as a loading control for WCE samples. Antibodies used for immunoprobing are indicated to the left of each panel. An asterisk "*" indicates heavy chain antibody. Source data are provided as a Source Data file.

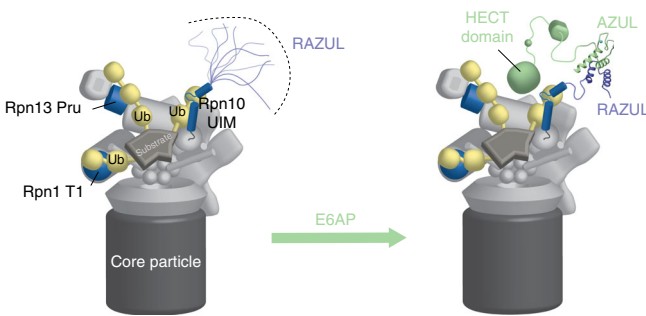

**Fig. 7 Model of hRpn10 RAZUL recruitment of E6AP to the 26S proteasome.** Proteasome substrate receptors Rpn1, Rpn10, and Rpn13 recognize ubiquitin chains (yellow) through their T1 site (blue), UIM region (blue), and Pru domain (blue), respectively. The Rpn10 C-terminal RAZUL is intrinsically disordered (left panel), but folds to form a 4-helix bundle with the AZUL domain of E3 ligase E6AP (green), thus recruiting E6AP to the proteasome (right panel). Zn, light blue.

conducive to ubiquitin recycling and the chain removal needed for substrate translocation into the proteolytic CP. Although it does not contribute a DUB-binding site, Rpn10 is proximal to the proteasomal DUB Rpn11[57,58]. Why would the proteasome be benefitted by physically linking a substrate receptor to a ubiquitin E3 ligase? One possibility is to spatially and temporally link its activity and its substrates with the proteasome. One model based on previous studies is that E6AP directly acts on proteasome subunits, considering that E6AP is reported to ubiquitinate proteasome components, including Rpn10[83,84], albeit at low levels[83–85]. Rpn10 is also ubiquitinated in yeast, which is proposed to regulate its presence at the proteasome[86,87] and affinity for ubiquitinated proteins[88]; however, yeast lack both RAZUL and E6AP (Supplementary Fig. 1a) suggesting Rpn10 ubiquitination has redundant mechanisms or is independent of E6AP. In agreement with this, we found no evidence of altered levels or a molecular weight increase indicative of ubiquitination for hRpn10 following E6AP knockdown in HCT116 cells (Fig. 3b), although it remains possible that such activity requires induction by a specific cellular event or is cell type specific.

Another possibility is that E6AP localization to the proteasome via hRpn10 RAZUL serves in the broader context to allow additional ubiquitin chains to be added to protein substrates, providing higher affinity and in turn, more efficient degradation. Various models have been proposed to explain how different chain lengths/linkages affect the degradation rate of proteins, with multiple short ubiquitin chains shown to have higher efficiency of degradation than a single long chain[89]. Multiple ubiquitin chains on a substrate may more readily enable multiplexed points of contact with receptor sites and associated/nearby DUBs around the degradation channel and thereby enable more efficient substrate translocation. As proteolysis by the proteasome requires a flexible initiation sequence to engage the ATPase ring of the RP[90–95], premature deubiquitination without substrate engagement with the ATPase ring can lead to release of a substrate prior to degradation. Substrates with multiple ubiquitin chains have higher affinity for the proteasome[89] and are deubiquitinated more slowly, favoring them for proteolysis by the proteasome, as observed by single molecule experiments[96].

E6AP interacting with the proteasome through the Rpn10 RAZUL would be ideally situated to modify substrates in this context, with proximity to substrates recruited by the Rpn10 UIMs or associated shuttle factors. This model is supported by our finding that E6AP knockdown or displacement from the proteasome reduces ubiquitin co-immunoprecipitating with the proteasome, despite largely unchanged ubiquitin levels in the WCE (Fig. 4c, d). It has also been demonstrated that ubiquitin chains consisting of K11/K48 branched chains are more efficiently degraded by the proteasome than K11-linked chains[97]. E6AP is known to catalyze K48 linkages[98–101], however its specificity has not been studied extensively. It is possible that E6AP could add K48-linked ubiquitin (or other linkages, as yet to be determined) to existing K11-linked chains, in order to enhance degradation of substrates. A role for ubiquitin chain remodeling has been proposed for UBE3C[102], which is the first ubiquitin ligase reported to physically interact with the proteasome. The UBE3C binding site is likely somewhere in the proteasome base subcomplex[102,103]; however, its location has yet to be elucidated, and its recruitment to the proteasome appears to be assisted by structurally impaired substrate[85].

As human E6AP isoform 3 localizes to the nucleus in an hRpn10- and AZUL-dependent manner[44], it may be that the E6AP's function in the nucleus is primarily related to its association with hRpn10. By contrast, the decreased affinity for E6AP when RAZUL is phosphorylated at Y326 may imply that certain cellular contexts require less E6AP associated with the proteasome (Fig. 7). For example, the identification of Y326 phosphorylation in Jukat cells (immortalized human T lymphocyte) may be significant given that immune cell activation is tightly regulated by ubiquitination[104]. E6AP has been reported to interact with Lck and Blk[105], which are immune cell-specific tyrosine kinases at the plasma membrane involved in T cell receptor (TCR) or B cell antigen receptor (BCR) mediated activation, respectively. It is conceivable that upon TCR or BCR activation, an immune cell-specific kinase is recruited to hRpn10 via E6AP, and that subsequent phosphorylation of hRpn10 leads to the reduction of E6AP at the proteasome, and in turn, delayed degradation of receptor components, allowing an elongated timeframe of activation. This mode of regulation could allow a fine-tuning of receptor activation by regulating E6AP activity at the proteasome in a very specific cellular location and context.

While future studies will elucidate the intricacies of E6AP function at the proteasome, our findings provide insight into the uniqueness of E6AP as an E3 ligase with a dedicated binding site at the proteasome and protein abundance that correlates with proteasome substrate receptor hRpn10. Our finding that E6AP binds a domain of hRpn10 not present in yeast in combination with our discovery that E6AP levels depend on hRpn10 suggests that these two proteins evolved together to have linked functions, underscoring the importance of E6AP at the proteasome. These results provide a new foundation towards understanding the role of E6AP in its associated disease states.

## Methods

**Protein sample preparation for biophysics experiments.** Human Rpn10[305–377] was subcloned between the EcoRI and XhoI restriction sites of the pGEX-6P-3 vector (GE Healthcare 28954651) in frame with an N-terminal glutathione S-transferase (GST) and a PreScission protease cleavage site, by using pET11d vector containing His$_6$-hRpn10[full-length] (a gift from Dr Fumio Hanaoka) and the appropriate primer pairs (Supplementary Table 1). The plasmid was transformed into E. coli strain Rosetta 2 (DE3) (MilliporeSigma 71400) with selection by ampicillin and chloramphenicol. The transformed cells were grown at 37 °C to OD$_{600}$ of 0.5–0.6 and protein expression induced overnight at 17 °C by addition of 0.4 mM IPTG. The cells were harvested, frozen in liquid nitrogen and stored at −80 °C for ~4 h, followed by resuspension in buffer 1 (50 mM Tris at pH 7.2, 300 mM NaCl, 5 mM DTT) supplemented with protease inhibitor cocktail tablets (Roche Diagnostics 11836153001). The cells were lysed by sonication and spun down at 27,000 g for 30 min, after which the supernatant was incubated with pre-washed glutathione sepharose resin (GE Healthcare 17075605) for 3 h. After extensive washing in buffer 1, hRpn10[305–377] was either separated from GST and the resin by cleaving with PreScission protease (GE Healthcare 27084301) in buffer 2 (10 mM MOPS at pH 6.5, 50 mM NaCl, 5 mM DTT, and 10 μM zinc sulfate), or eluted from the resin with the GST-tag intact by using buffer 3 (50 mM Tris at pH 7.2, 50 mM NaCl, 5 mM DTT, 20 mM glutathione). Further purification was achieved by size exclusion chromatography on an FPLC system ÄKTA pure (GE Healthcare) using a HiLoad 16/600 Superdex 75 (for samples with no GST-tag) or Superdex 200 (for GST-tagged protein) prep grade column in buffer 2 or 3.

E6AP AZUL, spanning amino acids 24 to 87, was subcloned between the NdeI and SacI restriction sites of the pET28a(+) vector (MilliporeSigma 69864) in frame with an N-terminal polyhistidine tag and a thrombin cleavage site, by using p4054 HA-E6AP isoform II (Addgene plasmid #8658 gifted from Dr Peter Howley)[106] and the appropriate primer pairs (Supplementary Table 1). The plasmid was transformed into E. coli strain BL21 (DE3) (Thermo Fisher Scientific C600003) with kanamycin selection. The transformed cells were grown at 37 °C to OD$_{600}$ of 0.5–0.6 and protein expression induced at 17 °C overnight by 0.4 mM IPTG. At the time of induction, zinc sulfate was added to a final concentration of 20 μM. The cells were harvested, frozen in liquid nitrogen and stored at −80 °C for ~4 h, followed by resuspension in buffer 4 (10 mM MOPS at pH 7.2, 300 mM NaCl, 5 mM 2-mercaptoethanol, 10 μM zinc sulfate) supplemented with EDTA-free protease inhibitor cocktail tablets (Roche Diagnostics 11836170001). The cells were lysed by sonication and spun down at 27,000 g for 30 min. The supernatant was incubated with pre-washed Ni-NTA agarose resin (Qiagen 30230) for 1 h. After extensive washing in buffer 4, E6AP AZUL was separated from the His-tag and the resin by cleaving with thrombin in buffer 5 (10 mM MOPS at pH 6.5, 50 mM NaCl, 5 mM 2-mercaptoethanol, and 10 μM zinc sulfate). Further purification was achieved by size exclusion chromatography on an FPLC system ÄKTA pure (GE Healthcare) using a HiLoad 16/600 Superdex 75 prep grade column in buffer 5.

hRpn10[196–377] was subcloned between the NdeI and EcoRI restriction sites of the pET14b vector (MilliporeSigma 69660) in frame with an N-terminal polyhistidine tag and a thrombin cleavage site, by using pET11d vector containing His$_6$-hRpn10[full-length] (a gift from Dr Fumio Hanaoka) and the appropriate primer pairs (Supplementary Table 1). pET11d expressing His$_6$-hRpn10[full-length] was a gift from Dr Fumio Hanaoka and His$_{10}$-hRpn10[196–306] was subcloned into pET26b[12,51]; all plasmids were validated by standard sequencing (Macrogen). N-terminal His-tagged hRpn10[full-length], hRpn10[196–306], and hRpn10[196–377] were expressed from E. coli strain BL21 (DE3) (Thermo Fisher Scientific C600003) and purified in an identical manner as N-terminal His-tagged E6AP AZUL, but eluted from the resin with the His-tag intact by using elution buffer 6 (10 mM MOPS at pH 7.2, 50 mM NaCl, 5 mM 2-mercaptoethanol, and 250 mM imidazole), and further purified by size exclusion chromatography on an FPLC system ÄKTA pure (GE Healthcare) with a HiLoad 16/600 Superdex 75 prep grade column in buffer 5.

$^{15}$N ammonium chloride, $^{13}$C glucose, and $^2$H$_2$O were used for isotope labeling. All NMR samples were validated by mass spectrometry. 26S proteasome (human) was purchased (Enzo Life Sciences, Inc. BML-PW9310).

**Peptide synthesis.** hRpn10[322–366] peptide without or with Y326 phosphorylated was synthesized on a Liberty Blue Microwave peptide synthesizer (CEM Corporation) using Fmoc chemistry. To avoid oxidation, Met residues in the sequence of HVR were substituted by isosteric norleucine. The following modifications were introduced to the published protocol for high efficiency peptide synthesis[107]. The coupling with N,N′-diisopropylcarbodiimide (DIC)/ethyl 2-cyano-2-(hydroxyimino)acetate (OXYMA) was performed for 4 min at 90 °C for all residues except Cys and His, for which the reaction was carried out for 10 min at 50 °C. Removal of the Fmoc group was conducted at 90 °C for 2 min for sequences containing no Cys

or Asp. All deprotection cycles after Asp and Cys were conducted at room temperature to avoid racemization and aspartimide formation. Low loading Rink Amide MBHA resin (Merck) was used for the synthesis of amidated peptides and Wang resins were used for the synthesis of peptides with free carboxy termini. The peptides were cleaved from the resin and deprotected with a mixture of 90.0% (v/v) trifluoroacetic acid (TFA) with 2.5% water, 2.5% triisopropyl-silane, 2.5% 2,2′-(ethylenedioxy)diethanethiol, and 5% thioanisol. Peptides were purified on a preparative (25 mm × 250 mm) Atlantis C3 reverse phase column (Agilent Technologies) in a 90 min gradient of 0.1% (v/v) TFA in water and 0.1% TFA in acetonitrile, with a 10 mL min$^{-1}$ flow rate. The fractions containing peptides were analyzed on an Agilent 6100 LC/MS spectrometer with the use of a Zorbax 300SB-C3 PoroShell column and a gradient of 5% acetic acid in water and acetonitrile. Fractions that were more than 95% pure were combined and freeze dried.

**GST-pulldown experiment and sample preparation for mass spectrometry.** HCT116 and 293T cells were purchased from American Type Culture Collection (ATCC CCL-247 and CRL-3216). 293T cells were grown in DMEM (Gluta-MAX$^{TM}$-1 with 4.5 g L$^{-1}$ D-glucose and without sodium pyruvate, Thermo Fisher Scientific) and HCT116 cells were grown in McCoy's 5A modified medium (ATCC 30-2007), with both media supplemented with 10% fetal bovine serum (Atlanta Biologicals, Inc. S12450). Cells were grown in a 37 °C humidified atmosphere of 5% CO$_2$. Cells were collected, washed twice with PBS, and lysed in 1% Triton-TBS buffer (50 mM Tris at pH 7.5, 150 mM NaCl, and 1 mM PMSF), supplemented with protease inhibitor cocktail (Roche Diagnostics 11836153001). Total protein concentration in the lysate was determined by a Pierce bicinchoninic acid protein assay kit (Thermo Fisher Scientific 23225). Overall, 2 nmol of purified GST-tagged hRpn10$^{305–377}$ or GST protein (Thermo Scientific 20237) was added to 20 μL of pre-washed glutathione sepharose resin for 3 h, washed once with buffer 7 (50 mM Tris at pH 7.5, 150 mM NaCl, 2 mM DTT, and 1.0% (v/v) Triton-X-100), and the resin next incubated with 1.2 mL of cell lysate for 3 h. Unbound protein was removed by washing three times in buffer buffer 7, after which resin-bound proteins were eluted, subjected to electrophoresis on a 4–12% NuPAGE Bis-Tris gel (Thermo Fisher Scientific NP0322), and visualized by Coomasie staining.

For each of the six lanes (GST-hRpn10$^{305–377}$ or GST protein incubated with 293T or HCT116 cell lysate), the region above 51 kDa was cut into 12 bands that were placed individually into 1.5 mL eppendorf tubes. Each gel band was then further cut into small pieces and destained by 50% acetonitrile/25 mM NH$_4$HCO$_3$ at pH 8. After removal of the organic solvent, gel pieces were dried by vacuum centrifugation for 1 h. Trypsin (20 ng μL$^{-1}$) in 25 mM NH$_4$HCO$_3$ at pH 8, was added to each sample (50 μL) and incubated on ice for 1 h. A total of 25 mM ammonium bicarbonate were added to completely saturate the bands for overnight incubation at 37 °C. Peptides were extracted in 70% acetonitrile and 5% formic acid using bath sonication and supernatant solutions were dried in a speed vacuum. Samples were desalted utilizing Pierce C18 spin columns (Thermo Fisher Scientific), dried, and resuspended in 0.1% TFA prior to mass spectrometry analysis.

**Mass spectrometry.** Peptides were analyzed on a Q Exactive Hybrid Quadrupole-Orbitrap mass spectrometer (Thermo Fisher Scientific). The desalted tryptic peptide was loaded onto an Acclaim PepMap 100 C18 LC column (Thermo Fisher Scientific) utilizing a Thermo Easy nLC 1000 LC system (Thermo Fisher Scientific) connected to the Q Exactive mass spectrometer. The peptides were eluted with a 5–48% gradient of acetonitrile with 0.1% formic acid over 55 min with a flow rate of 300 nL min$^{-1}$. The raw MS data were collected and analyzed in Proteome Discoverer 2.2 (Thermo Fisher Scientific) with Sequest HT software and was searched against the Human Proteome database.

**His pull-down assays.** His pull-down assays were performed as described previously[11,108]. Briefly, 2 nmol of purified His-tagged hRpn10$^{full-length}$, hRpn10$^{196–377}$, or hRpn10$^{196–306}$ was added to 10 μL of pre-washed Ni-NTA agarose resin (QIAGEN 30230) for 2 h and washed once with buffer 5. The resin was then incubated with 200 pmol of E6AP (UBPBio K1411) for 1 h and unbound protein removed by extensive washing with buffer 5. Resin-bound proteins were eluted and subjected to SDS-PAGE followed by visualization with Coomasie staining.

For pull-down assays testing whether E6AP impacts the affinity of the 26S proteasome for ubiquitin chains, 70 pmol of a commercially available mixture product of His$_6$-tagged, non-cleavable K48-linked Ub$_2$/Ub$_4$ (UBPBio D1701) was added to pre-washed Ni-NTA agarose resin (QIAGEN 30230) for 1 h and washed once with buffer 8 (50 mM Tris, 50 mM NaCl, 5 mM ATP, 5 mM MgCl$_2$, 5 mM 2-mercaptoethanol, 10 μM zinc sulfate at pH 7.5). The resin was then incubated with 10 pmol of human erythrocyte 26S proteasome (Enzo Life Sciences, Inc., BML-PW9310) alone, 10 pmol of 26S proteasome with equimolar of E6AP (UBPBio K1411), or 10 pmol of E6AP alone for 1 h and unbound protein removed by extensive washing with buffer 8. A total of 10 pmol of human 26S proteasome or E6AP were added to Ni-NTA agarose resin, incubating for 1 h and washing extensively with buffer 8, as negative controls. Resin-bound proteins were eluted and subjected to SDS-PAGE followed by visualization with Coomasie staining.

**ITC and SPR binding affinity experiments.** ITC was performed at 25 °C on a MicroCal iTC200 system. hRpn10$^{305–377}$, E6AP AZUL, and hRpn10$^{322–366}$ peptide without or with Y326 phosphorylated were dialyzed extensively against buffer 2. Eighteen 2.1 μL aliquots of 0.462 mM E6AP AZUL were injected at 1000 rpm into a calorimeter cell (volume 200.7 μL) that contained 0.0405 mM hRpn10$^{305–377}$. For measuring interaction between hRpn10$^{322–366}$ peptides and E6AP AZUL, eighteen 2.1 μL aliquots of 0.110 mM E6AP AZUL were injected at 1000 rpm into a calorimeter cell (volume 200.7 μL) that contained 0.01 mM hRpn10$^{322–366}$ without or with Y326 phosphorylated. Blank experiments were performed by replacing protein samples with buffer and this blank data was subtracted from the experimental data during analysis. The integrated interaction heat values were normalized as a function of the molar ratio of E6AP AZUL to hRpn10$^{305–377}$ or to hRpn10$^{322–366}$ peptides, and the data were fit with MicroCal Origin 7.0 software. Binding was assumed to be at one site to yield the binding affinity $K_a$ (1/$K_d$), stoichiometry, and other thermodynamic parameters.

SPR experiments were recorded for GST-tagged hRpn10$^{305–377}$ and E6AP AZUL with a Biacore T200 system (GE Healthcare). Utilizing the GST capture kit, 4000 RU of anti-GST antibody was covalently immobilized on a CM5 chip via amine coupling. GST-tagged hRpn10$^{305–377}$ was then added to FC2 to a final response of 800 RU. As a negative control, GST was added to FC1 to the same response. E6AP AZUL was prepared in degassed, filtered HBS-P+ (GE Healthcare) buffer with 10-μM zinc sulfate. Single cycle kinetic experiments were performed using five injections (30 μL min$^{-1}$) of increasing concentration of protein (5–250 nM) passed over the sensor chip for 150-s association, followed by a 420-s dissociation. The experiments were repeated in triplicate. Buffer and reference subtracted kinetic constants ($k_{on}$ and $k_{off}$) and binding affinities ($K_d$) were determined utilizing the Biacore T200 evaluation software (GE Healthcare).

**CD experiments.** Far-UV range CD spectra (240–190 nm) of 10 μM hRpn10$^{305–377}$, 10 μM E6AP AZUL, the mixture of 10 μM hRpn10$^{305–377}$ and 10 μM E6AP AZUL, and buffer 9 (10 mM MOPS, 10 mM NaCl, 10 mM DTT, 10 μM zinc sulfate at pH 6.5, as a control) were recorded on a Jasco J-1500 CD spectrometer (Tokio, Japan) using a quartz cuvette with 1.0 mm path length and temperature controlled at 25 ± 0.1 °C. All spectra were collected continuously at a scan speed of 20 nm/min and averaged over accumulation of three spectra. The buffer spectrum was subtracted from the protein spectra during data analyses. The molar ellipticity θ (in deg cm$^2$ dmol$^{-1}$) was calculated from the measured machine units m° in millidegrees at wavelength λ using the Eq. 1.

$$\theta = \frac{m°}{(10*C*L)} \qquad (1)$$

C is the concentration of the sample in mol L$^{-1}$ and $L$ is the path length of the cell (cm). Secondary structure analysis was conducted with the program CONTIN[109] at DichroWeb server[110,111] by using the reference dataset SMP180 (190–240 nm)[112].

**Cell culture, plasmids, siRNAs, and transfections.** All cell lines were grown in McCoy's 5A modified medium (Thermo Fisher Scientific 16600082), with 10% fetal bovine serum (Atlanta Biologicals, Inc. S12450) at 37 °C and 5% CO$_2$. The HCT116 cell line was purchased from the ATCC (CCL-247). Myc-tagged hRpn10 constructs were generated by Genscript by inserting synthesized hRpn10 (NM_002810.2) full-length coding sequence or DNA encoding for residues 1–306 between the KpnI and XhoI restriction sites of pcDNA3.1(+)-N-Myc. The resulting constructs included an N-terminal myc tag with a Gly-Thr linker between the myc tag and hRpn10 due to the KpnI site. Myc-tagged hRpn10$^{305–377}$ expressing plasmid was generated by PCR using pcDNA3.1(+)-N-Myc hRpn10$^{full-length}$ as the template and appropriate primer pairs (Supplementary Table 1), and the amplified DNA was subcloned between the KpnI and XhoI restriction sites of pcDNA3.1(+)-N-Myc vector (Genescript). pcDNA3.1(+)-N-Myc hRpn10 Y326F and pcDNA3.1(+)-N-Myc hRpn10$^{1–364}$ were generated by the QuikChange Site-Directed Mutagenesis Kit (Stratagene) using pcDNA3.1(+)-N-Myc hRpn10$^{full-length}$ plasmid as the template and appropriate primer pairs (Supplementary Table 1). All reconstructed plasmids were validated by standard DNA sequencing (Macrogen). p4054 HA-E6AP isoform II was purchased from Addgene (plasmid #8658), originally constructed in the laboratory of Peter Howley[106]. Plasmid transfections (48 h) were carried out with Lipofectamine 3000 (Thermo Fisher Scientific). siRNA transfections (72 h) were performed with Lipofectamine RNAiMAX (Thermo Fisher Scientific). Conditions labeled "mock" received only RNAiMAX, conditions labeled "scramble" were transfected with ON-TARGETplus non-targetting siRNA #2 (Dharmacon D-001810-02). hRpn10 and E6AP siRNAs used were ON-TARGETplus siRNAs (Dharmacon 011365 and 005137, respectively). Where only one siRNA from the set was used, the siRNAs were 011365-05 and 005137-05.

**CRISPR-Cas9 generation of ΔRAZUL.** Candidate sgRNAs flanking the starting portion of RAZUL were identified by using sgRNA Scorer 2.0[113]. Oligonucleotides for each sgRNA were annealed and ligated into a vector carrying Cas9-2A-mCerulean. The Cas9-2A-mCerulean was generated by digesting the pX458 backbone and replacing eGFP with mCerulean. pX458 was a gift from Feng Zhang (Addgene plasmid #48138; http://n2t.net/addgene:48138; RRID:Addgene_48138)[114]. Plasmid donor used as template for homology-directed repair was generated using isothermal

assembly of the left and right homology arms with the P2A-puromycin cassette. The left and right homology arms were amplified using PCR from 293T genomic DNA and mutations were introduced in the PAM sequence of each target site to prevent editing of the dsDNA donor. Sequences of all oligos used are provided in Supplementary Table 2. HCT116 cells were co-transfected with the Cas9-2A-mCerulean constructs in combination with the donor construct. Two days post transfection, cells were split into a fresh six-well plate and grown in media with puromycin for 2 weeks. A few cell colonies were visible after 2 weeks that were picked using sterile tips and added to a 24-well plate with fresh media. Individual clones were then analyzed by immunoblotting using hRpn10 antibody to identify clones containing the truncation. Clones 13 and 14 were selected based on this analysis, and genomic DNA isolated from clones 13 and 14 were used in PCR amplification using primers outside the homology arms (Supplementary Table 1). PCR fragments of the expected size were generated, and PCR using one primer outside the homology arm and one to puromycin validated insertion of the donor sequence. PCR-amplified genomic DNA was cloned into the pCR™4-TOPO® vector using a TOPO™ TA Cloning™ Kit (Life Technologies K457501) and sequence verified.

**Cell lysis and immunoprecipitations.** All cells were washed twice in cold PBS (Thermo Fisher Scientific) prior to harvesting. Cells used for Rpt3 IP were harvested on ice in 1% Triton X-100 buffer (1% Triton X-100, 50 mM Tris pH 7.5, 150 mM NaCl, 1 mM PMSF, 5 μg mL$^{-1}$ Pepstatin A, and Roche EDTA-free protease inhibitor cocktail). Cell extracts were centrifuged for 15 min at 4 °C and 20,000 g, and the supernatant was isolated. IPs were performed overnight at 4 °C with 1–1.5 mg total protein lysate, using 4 μL Rpt3 antibody (abcam ab140515) per condition. Protein G sepharose beads (GE healthcare 17-0618-01) were added to IPs for 3 h, and precipitates were washed 5–7 times with 1% Triton X-100 buffer. Immune complexes were heated to 95 °C for 10 min in denaturing sample buffer prior to subjection to SDS-PAGE.

Cells used for myc-trap IPs were harvested on ice in 0.5% NP-40 lysis buffer (0.5% Nonidet P-40, 50 mM Tris pH 7.5, 150 mM NaCl, 1 mM PMSF, 5 μg mL$^{-1}$ Pepstatin A, 10 mM sodium pyrophosphate, 10 mM NaF, 1 mM Na$_3$VO$_4$, and Roche EDTA-free protease inhibitor cocktail). Cell extracts were placed on ice for 30 min with extensive pipetting every 10 min. Extracts were centrifuged at 20,000 × g for 30 min at 4 °C. Supernatants were diluted 1:1 with myc-trap dilution/wash buffer (50 mM Tris pH 7.5, 150 mM NaCl, 1 mM PMSF, 5 μg mL$^{-1}$ Pepstatin A, 10 mM sodium pyrophosphate, 10 mM NaF, 1 mM Na$_3$VO$_4$, and Roche EDTA-free protease inhibitor cocktail), and 1–1.5 mg total protein lysate was incubated with 25 μL myc-trap agarose (nanobody coupled) beads (chromotek) overnight at 4 °C. Nanobody-myc complexes were washed five times with myc-trap dilution/wash buffer and heated to 95 °C for 10 min in denaturing sample buffer prior to subjection to SDS-PAGE.

In experiments not involving immunoprecipitation, cells were either harvested in 1% Triton X-100 buffer or 1% NP-40 buffer (1% Nonidet P-40, 25 mM Tris pH 7.2, 137 mM NaCl, 10% glycerol, 1 mM DTT, 5 mg mL$^{-1}$ Pepstatin A, 1 mM PMSF, and Roche protease inhibitor cocktail). Lysates were centrifuged at 20,000 × g and 4 °C for 15 min and supernatants were isolated for immunoblotting.

**SDS-PAGE, immunoblots, and antibodies.** Protein lysates were subjected to SDS-PAGE on 4–12% NuPAGE Bis-Tris gels (Thermo Fisher Scientific NP0322) using MOPS SDS running buffer (Thermo Fisher Scientific NP0001), except in the case of Fig. 4b, d, where MES SDS running buffer (Thermo Fisher Scientific NP002) was used to achieve better resolution of myc-RAZUL. Proteins were transferred to 0.45 μm PVDF membranes (Thermo Fisher Scientific LC2005) using NuPAGE transfer buffer (Thermo Fisher Scientific NP00061) supplemented with 10% methanol. Following transfer, membranes were blocked with 5% milk in tris-buffered saline with 1% tween 20 (TBS-T). Blocked membranes were incubated with primary antibodies (diluted in 5% milk in TBS-T) overnight. Membranes were washed five times in TBS-T and incubated with HRP-conjugated secondary antibodies (diluted in 5% milk in TBS-T) for 2 h. Following another set of five washes, blots were developed using HyGlo quickspray chemiluminescent HRP detection reagent (Denville Scientific Inc. E2400) and HyBlot CL Autoradiography Film (Denville Scientific Inc. E3018). Primary antibodies used were: β-actin (Cell Signaling Technologies 4970, 1:3,000), Cyclophilin B (Abcam ab178397, 1:10,000), E6AP (MilliporeSigma E8655, 1:1,000), HA-tag (MilliporeSigma H6908, 1:1,000), myc-tag (Cell Signaling Technologies 2278, 1:1,000), Rpn10 (Novus Biologicals NBP2-19952, 1:1,000), Rpn2 (Bethyl Laboratories A303-851A, 1:10,000), Rpn13 (Abcam ab140515, 1:5,000), Rpn11 (Cell Signaling Technologies 4197, 1:1,000), Rpn8 (abcam ab140428, 1:1,000), and ubiquitin (MilliporeSigma MAB1510, 1:1,000 or Cell Signaling Technologies 3936, 1:1,000). Secondary antibodies used were: Rabbit HRP (Life technologies A16110, 1:50,000) and mouse HRP (MilliporeSigma A9917, 1:5,000). For blots in which a protein ran close to the heavy chain in an IP condition, Native Rabbit HRP (MilliporeSigma R3155, 1:1,000) was used.

**NMR samples and experiments.** Three NMR samples were prepared, including (1) 0.5 mM $^{15}$N, $^{13}$C, 35% $^2$H-labeled Rpn10$^{305–377}$ mixed with unlabeled E6AP AZUL at 1.5-fold molar excess; (2) 0.5 mM $^{15}$N, $^{13}$C E6AP AZUL mixed with unlabeled Rpn10$^{305–377}$ at 1.5-fold molar excess; (3) 0.5 mM $^{15}$N, $^{13}$C, 70%

$^2$H-labeled Rpn10$^{196–377}$. For assignment of Rpn10$^{305–377}$ in the E6AP-bound state, or E6AP in the Rpn10$^{305–377}$-bound state, 2D $^1$H-$^{15}$N HSQC and $^1$H-$^{13}$C HSQC, 3D HNCACB/CBCA(CO)NH, HNCO/HN(CA)CO, HCCH-TOCSY, CCH-TOCSY, $^{15}$N (120 ms mixing time) and $^{13}$C (80 ms mixing time) edited NOESY-HSQC spectra were recorded on samples 1 and 2. These experiments were also recorded on sample 3 to obtain free state assignments for Rpn10$^{196–377}$. Unambiguous intermolecular distance constraints were obtained by using 3D $^{13}$C-half-filtered NOESY experiments (100 ms mixing time) recorded on samples 1 and 2. Chemical shift assignment of E6AP AZUL in the free state was available from a previous study[45].

All NMR experiments were conducted at 25 °C in buffer 10 (10 mM MOPS at pH 6.5, 50 mM NaCl, 5 mM DTT, 10 μM zinc sulfate, 1 mM pefabloc, 0.1% NaN$_3$, and 5% $^2$H$_2$O/95% $^1$H$_2$O), except for 2D $^1$H-$^{13}$C HSQC, 3D HCCH-TOCSY, CCH-TOCSY, $^{13}$C-edited NOESY-HSQC and $^{13}$C-half-filtered NOESY experiments, which were acquired on samples dissolved in $^2$H$_2$O. Spectra were recorded on Bruker AvanceIII 600, 700, 800, or 850 MHz spectrometers equipped with cryogenically cooled probes.

All NMR data processing was performed with NMRpipe[115] and spectra were visualized and analyzed with XEASY[116]. Secondary structure was assessed by comparing chemical shift values of Cα and C' atoms to random coil positions to generate a CSI[117] and also by the TALOS+ program[118].

**NMR titration experiments.** $^1$H, $^{15}$N HSQC experiments were recorded on 0.2 mM $^{15}$N-labeled samples (hRpn10$^{305–377}$ or E6AP AZUL) with increasing molar ratio of unlabeled ligand (E6AP AZUL or hRpn10$^{305–377}$), as indicated. The amide nitrogen and hydrogen chemical shift perturbations were mapped for each amino acid according to Eq. 2.

$$CSP = (0.2\Delta\delta_N^2 + \Delta\delta_H^2)^{1/2} \qquad (2)$$

$\Delta\delta_H$, change in amide proton value (in parts per million); $\Delta\delta_N$, change in amide nitrogen value (in parts per million).

**1D $^{13}$C-edited, $^1$H NMR experiments.** Three NMR samples were prepared in buffer 11 (50 mM d$^{11}$-Tris at pH 7.5, 50 mM NaCl, 5 mM MgCl$_2$, 1.5 mM ATP-γS, 10 μM zinc sulfate, 2 mM DTT, 0.5 mM pefabloc and 5% $^2$H$_2$O / 95% $^1$H$_2$O), including 0.3 μM of free $^{13}$C labeled AZUL domain, 0.3 μM of $^{13}$C labeled AZUL domain mixed with equimolar unlabeled RAZUL, and 0.3 μM of $^{13}$C labeled AZUL domain mixed with equimolar human 26 S proteasome (Enzo Life Sciences, Inc., BML-PW9310). 200,000 1D traces of a $^1$H, $^{13}$C HSQC experiment[119,120] were averaged for each sample at 25 °C and 850 MHz with a cryogenically cooled probe.

**Structure calculation of the hRpn10 RAZUL: E6AP AZUL complex.** XPLOR-NIH 2.50[121] was used on a Linux operating system to solve the complexed structure by using NOE and hydrogen bond constraints as well as backbone ϕ and ψ torsion angle constraints derived from TALOS+[118] (Table 1). Hydrogen bonds were generated by using secondary structure assignments and NOE connectivities with defined distances from the acceptor oxygen to the donor hydrogen and nitrogen of 1.8–2.1 and 2.5–2.9 Å, respectively. Hydrogen bonds restraints were not included in the initial calculation but were in the final round of structure calculations. When calculating the structures of hRpn10 RAZUL:E6AP AZUL, intermolecular distance constraints determined from the 3D $^{13}$C-half-filtered NOESY experiments were used, in addition to intramolecular constraints for hRpn10 RAZUL and E6AP AZUL that were generated from $^{15}$N or $^{13}$C NOESY spectra acquired on the complexes (Table 1). The complexed structures were calculated from 50 linear starting structures of hRpn10 RAZUL and E6AP AZUL molecules, which were subjected to 2000 steps of initial energy minimization to ensure full spatial sampling and appropriate coordinate geometry. The structures were next confined according to the inputted data by subjecting them to 55,000 simulated annealing steps of 0.005 ps at 3000 K, followed by 5000 cooling steps of 0.005 ps. 5000 steps of energy minimization were applied to produce the final structures, which were recorded as coordinate files. The resulting structures had no distance or dihedral angle violation greater than 0.3 Å or 5°, respectively. The 15 lowest energy structures were chosen for visualization and statistical analyses. Structure evaluation was performed with the program PROCHECK-NMR[122]; the percentage of residues in the most favored, additionally allowed, generously allowed and disallowed regions were 95.4, 4.5, 0.1, and 0.0, respectively. Visualization was performed with MOLMOL[123] and PyMOL (PyMOL Molecular Graphics System, http://www.pymol.org). The electrostatic surface of E6AP AZUL was generated by the Poisson-Boltzmann (APBS) method[124,125].

**Reporting summary.** Further information on research design is available in the Nature Research Reporting Summary linked to this article.

## Data availability

Atomic coordinates for RAZUL:AZUL have been deposited in the Protein Data Bank (PDB) with accession number 6U19. Chemical shift assignments have been deposited in the Biological Magnetic Resonance Data Bank (BMRB) with accession number 27875. The source data underlying Figs. 1b, e, f, 2a–c, 3a–d, 4a–e, 5c, d, and 6g and

Supplementary Figs. 2a, c, e, 3, 4a, and 6b are provided as a Source Data file. Other data are available from the corresponding authors upon reasonable request.

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

## Acknowledgements

We thank Charles D. Schwieters (CIT, NIH) for support with XPLOR-NIH and Janusz Koscielniak for technical assistance with the NMR facility. This work was supported by the Intramural Research Program through the CCR, NCI, NIH (1 ZIA BC011490 and the CCR FLEX program).

## Author contributions

G.R.B. performed molecular biology and cell biology studies. X.C. performed CD, NMR, and structure calculations. R.C. designed and generated plasmids for making *ΔRAZUL* cells. M.O. and T.A. performed SPR. C.J. and T.A. performed mass spectrometry analysis. D.L.E. prepared cell lysate for GST pull-down. V.S. made and isolated clones of *ΔRAZUL* cells. S.G.T. performed ITC. N.I.T. synthesized peptides. K.J.W. designed experiments. G.R.B., X.C., and K.J.W. wrote/edited the manuscript.

## Competing interests

The authors declare no competing interests.
