## [Peer Review File · Nature Communications]

Reviewers' comments:

Reviewer #1 (Remarks to the Author):

Overview:

-E6AP/UBE3A binds the proteasome substrate receptor hRpn10/PSMD4/S5a tightly with a 12 nM dissociation constant. This novel finding localizes a physiologically important E3 directly to the proteasome. The functional outcome of this binding is not yet known.

Summary of Results:

-using NMR, it is shown that the C-terminal sequence of hRpn10 does not bind Ub (or its UIMs), but binds the E6AP (E3 Ub ligase) AZUL domain to form a four helix bundle in the complex.

Pull downs (GST-hRpn10 305-377) using 293T or colorectal carcinoma cells revealed E6AP as the binding partner, as identified by mass spectrometry and proteomics.

Pull downs (Ni-NTA; His-hRpn10 full length, N-terminal only, or C-terminal only) with epithelial or cancer cells were conducted and E6AP was identified as a binding partner. Only the C-terminal region of hRpn10 bound full length E6AP.

Complementary ¹⁵N NMR experiments for E6AP AZUL and hRpn10305-377 show significant chemical shift changes and indicate binding. ITC and SPR studies indicate a strong interaction with a dissociation constant around ten nanomolar.

-Endogenous studies.

co-IP pulls down E6AP using myc-hRpn10 and HA-E6AP co-precipitates full length hRpn10, C-terminal deletions of hRpn10 do not interact with E6AP.

CRISPR knockouts (siRNA leads to proteasome defects, Fig. 2A): generated hRpn10 lacking an AZUL domain to circumvent proteasome defects (rather than deleting all of hRpn10). E6AP protein levels are correlated with the level of hRpn10 delta-RAZUL. The CRISPR knockouts were used subsequently used in immunoprecipitation experiments to show that the RAZUL domain localizes E6AP to the proteasome.

1D ¹³C NMR studies were used to show that the RAZUL:AZUL interaction remains the same at the proteasome.

NMR was used to determine a high quality solution structure of the RAZUL:AZUL interaction, the complex forms a four helix bundle.

Comments:

-on line 35 of the abstract, it is not clear what "this domain" is, need to specify which domain of hRpn10 is being discussed.

-Some of the same residues in figure S1B should be labeled as in figure 1c, to orient the reader (those residue numbers > residue 377).

-For Figure S2A, what band does E6AP corresponds to (<97 kDa?), and what are the other bands between 51 kDa and >97 kDa for 293T and HCT116 cells? Is this a proteolysis problem?

-Fig. 3c has been mis-labeled as Fig. 3d.

-It is difficult to interpret the closely spaced (mass), multiple bands in Fig. 1f, was there a problem with proteolysis or PTMs, protease inhibitors appear to have been used as appropriate?

-On line 164, the figure callout should be Fig. 2a.

-On lines 171-173, shouldn't this callout be Fig. 2b?

-some clarification would be helpful regarding the ¹³C NMR experiments. The binding of hRpn at the proteasome (affinity/structure/domains), and whether the domains are flexible in the context of binding to the proteasome should be discussed, and that 1:1 binding is likely a saturating amount. This would help to better interpret the changes in the observed NMR spectra (just line broadening for the complex binding the proteasome).

-the terminal carbonyl CSI value should be checked (Supp. Fig. 6B).

-the discussion regarding the function of localizing E6AP to the proteasome is speculative, but it is strongly implied that its role is to build mixed 11/48 chains at the proteasome to allow enhanced substrate degradation. It is just as likely that this mechanism simply leads to highly efficient degradation of substrates by localizing them directly to the proteasome. This would also require tight regulatory control of E6AP proteasome binding, and phosphorylation of Y326 might be involved. The E3 itself might also be auto-inhibited at the proteasome.

-If possible, some structural models involving E6AP and how it might be interacting with the proteasome, relevant domain dynamics when bound, and potential E3 regulatory mechanisms could enhance the paper, along the lines of Sailer, *Nat. Comm.* 9, 4441, 2018.

Reviewer #2 (Remarks to the Author):

In this article, Buel, Chen, et. al describe a previously unappreciated binding site for the E6AP/UBE3A E3 ubiquitin ligase encoded by proteasomal substrate receptor Rpn10. This binding site, termed RAZUL, demonstrates low nM affinity for E6AP. Using human cells in which the RAZUL domain has been deleted, the authors demonstrate reduced interaction between E6AP and Rpn10ΔRAZUL in cells, and that this results in reduced levels of E6AP associated with the proteasome. The authors determine the structure of the free RAZUL domain and the RAZUL domain in complex with the AZUL domain from E6AP. Upon interaction with AZUL, RAZUL undergoes a disordered-to-structured transition that results in formation of a four-helix bundle consisting of two helices each from AZUL and RAZUL. Finally, the authors demonstrate that phosphorylation of a tyrosine residue (reported in a mass spectrometry study) reduces the affinity of RAZUL for AZUL approximately 10-fold. Based on these observations, the authors propose that E6AP has a "privileged" position on the proteasome to modify substrates that associate with Rpn10.

Overall, the manuscript is thorough, with multiple lines of experimental evidence supporting each main conclusion. Importantly, this manuscript provides physiological context for a major evolutionary and structural difference between the highly studied fungal Rpn10 and the human ortholog. There are, however, some points that should be addressed to clarify and strengthen a key conclusion of the manuscript.

Major points

1. The authors conclude that E6AP has a privileged position at the proteasome via a dedicated binding site on Rpn10, where it modifies substrate ubiquitin chain architecture to regulate proteolysis. In support of this, the authors knock down E6AP with siRNA, and demonstrate reduced ubiquitin association with the proteasome. A simple and much more powerful experiment would be

to examine ubiquitin association +/- E6AP knockdown in the Rpn10 Δ RAZUL cells. If positioning of E6AP on Rpn10 (vs. ubiquitination of substrates by free E6AP or E6AP bound to some other portion of the proteasome) were truly what confers this privilege, then the E6AP knockdown should have little or no effect in the Rpn10 Δ RAZUL cells. Similarly, does overproduction of the RAZUL peptide as in Fig. 3b reduce ubiquitination on the proteasome?

2. Considering that Rpn10 is likely the major ubiquitin receptor on the proteasome, another potentially trivial explanation for the reduced ubiquitin associated with the proteasome in E6AP knockdown cells (Fig. 3) is that interaction with E6AP alters the affinity of Rpn10 for ubiquitinated substrates in the context of the proteasome. Is the interaction between proteasome-associated Rpn10 and ubiquitin/substrates similar when E6AP is knocked down?

3. Is there any evidence that substrate turnover is impaired or otherwise altered by loss of E6AP at the proteasome?

Minor points

1. As written, the abstract implies that Rpn10 Δ RAZUL cells display reduced ubiquitin associated with the proteasome. This was not directly tested (although see major point above).

2. p. 7: several references to figures are incorrect in the first two paragraphs.

3. Figure 3c is mislabeled as "d."

4. The authors indicate early in the manuscript that they have studied E6AP isoform 2; some (brief) commentary on whether the other isoforms of E6AP are likely to interact with Rpn10 in the same manner would be useful for naïve readers.

5. The sentence in line 206 contains a grammatical error (replace contributes with recruits?).

Reviewer #1 (Remarks to the Author): We greatly appreciate the careful reading and valuable suggestions made by Reviewer #1. We have made all corrections and integrated all suggestions including the addition of a functional model as Figure 7. All changes are indicated in the revised manuscript by using red font.

Comments:

-on line 35 of the abstract, it is not clear what “this domain” is, need to specify which domain of hRpn10 is being discussed.

We have now defined ‘this domain’ by replacement with ‘the E6AP-binding domain’.

-Some of the same residues in figure S1B should be labeled as in figure 1c, to orient the reader (those residue numbers > residue 377).

Thank you for the suggestion. In the new figure S1B, we have now labeled some residues that are also labeled in figure 1c; these are denoted by using grey italicized font.

-For Figure S2A, what band does E6AP corresponds to (<97 kDa?), and what are the other bands between 51 kDa and >97 kDa for 293T and HCT116 cells? Is this a proteolysis problem?

In the new Figure S2A, the band corresponding to the molecular weight of E6AP in the gel is now indicated with an asterisk. We agree that there is a prominent lower molecular weight band between 51 and 64 kDa that could potentially be a breakdown product. We did not repeat or try to optimize this experiment, as we focused on the E6AP discovery/validation. The mass spectrometry analyses did not yield a possible candidate for this band – offering support for it being a breakdown product.

-Fig. 3c has been mis-labeled as Fig. 3d.

Thank you for pointing this error out; we have now fixed this typo.

-It is difficult to interpret the closely spaced (mass), multiple bands in Fig. 1f, was there a problem with proteolysis or PTMs, protease inhibitors appear to have been used as appropriate?

In Fig. 1f, both E6AP and hRpn10 are expressed with tags, which cause the proteins to run slightly higher than the endogenous forms, which are also present in these cells. Therefore, two closely spaced bands are observable in these cases; we have now made this aspect clearer in the figure caption. For E6AP, which has an HA tag, the anti-HA antibody has a background band that we have now highlighted in both the IP and whole cell extract images with an asterisk to avoid confusion. In the case of hRpn10, we are expressing both a truncated form (runs around the 39 kDa marker) and a full-length form (runs around the 51 kDa marker) in cells that contain endogenous hRpn10. We do see a bit of a breakdown product from the exogenously expressed Rpn10 constructs, likely because the level of expression we are getting is higher than the endogenously expressed. However, since the amount of breakdown is very small in comparison to the level of intact protein, we do not think this breakdown product is affecting the results of the experiment.

-On line 164, the figure callout should be Fig. 2a.

Thank you for pointing this error out; we have now fixed this error.

-On lines 171-173, shouldn't this callout be Fig. 2b?

Yes, many thanks for pointing this error out, which we have now fixed.

-some clarification would be helpful regarding the ¹³C NMR experiments. The binding of hRpn at the proteasome (affinity/structure/domains), and whether the domains are flexible in the context of binding to the proteasome should be discussed, and that 1:1 binding is likely a saturating amount. This would help to better interpret the changes in the observed NMR spectra (just line broadening for the complex binding the

proteasome).

Thank you for drawing our attention to these omissions; we have now added these further explanations to the text and agree that the clarifications improve readability of the manuscript.

-the terminal carbonyl CSI value should be checked (Supp. Fig. 6B).

Yes, we noticed that the CSI value of K377 (the terminal carbonyl) is ~5 ppm for both RAZUL alone and when bound to AZUL. We have checked the spectra very carefully and confirmed that the chemical shift of C' for K377 is correct. In HNCACO and HNCOC spectra, we used 9.5 ppm for the carbon sweep width. To confirm that the chemical shift of K377 C' is not caused by spectral folding, we acquired the 2D proton-carbon plane of the HNCACO spectrum with a larger carbon sweep width of 20 ppm (please see Response Figure 1 below). The chemical shift value of the K377 C' remains at 181.42 ppm in both spectra. Generally, C' atoms of the C-terminal residue (which is not in an isopeptide bond) are shifted downfield (larger values). This trend has been observed previously; for example, the C' CSI value of the Rpt6 C-terminal residue (Ehlinger, A. *et al.* Conformational dynamics of the Rpt6 ATPase in proteasome assembly and Rpn14 binding. *Structure* **21**, 753-765). We have now noted this phenomenon in the figure caption for clarity.

Response Figure 1. 2D proton-carbon plane of an HNCACO spectrum for ^{15}N , ^{13}C , 35% ^2H -labeled RAZUL mixed with unlabeled AZUL. A carbon sweep width of 9.5 ppm (left panel) or 20 ppm (right panel) is used to demonstrate that K377 is not folded by a shorter sweep width. The chemical shift values for the K377 amide proton and C' are highlighted by blue dashed lines and labeled.

-the discussion regarding the function of localizing E6AP to the proteasome is speculative, but it is strongly implied that its role is to build mixed 11/48 chains at the proteasome to allow enhanced substrate degradation. It is just as likely that this mechanism simply leads to highly efficient degradation of substrates by localizing them directly to the proteasome. This would also require tight regulatory control of E6AP

proteasome binding, and phosphorylation of Y326 might be involved. The E3 itself might be also be auto-inhibited at the proteasome.

We agree with all of these points and did not mean to emphasize the mixed chain importance above the overall increase in affinity that would result by multi-ubiquitination or the tight spatial and temporal coupling between ubiquitination and proteasome activity for E6AP substrates, given its presence at the proteasome. We have now expanded our Discussion accordingly to better highlight the expected implications of E6AP localization to the proteasome.

-If possible, some structural models involving E6AP and how it might be interacting with the proteasome, relevant domain dynamics when bound, and potential E3 regulatory mechanisms could enhance the paper, along the lines of Sailer, Nat. Comm. 9, 4441, 2018.

We thank the Reviewer for this recommendation as we agree that inclusion of the model adds to readability, particularly for those not embedded in the field. We have made such a structural model and include it in the revised version of the manuscript as Fig. 7.

Reviewer #2 (Remarks to the Author): We thank Reviewer #2 for their enthusiasm towards our manuscript and for their much-valued suggestions/corrections. We have addressed all of the concerns and included in the manuscript two new figures (4d and 4e) based on the Reviewer's suggestions. All changes to the text of the manuscript are indicated in the revised version by red font.

Major points

1. The authors conclude that E6AP has a privileged position at the proteasome via a dedicated binding site on Rpn10, where it modifies substrate ubiquitin chain architecture to regulate proteolysis. In support of this, the authors knock down E6AP with siRNA, and demonstrate reduced ubiquitin association with the proteasome. A simple and much more powerful experiment would be to examine ubiquitin association +/- E6AP knockdown in the Rpn10 Δ RAZUL cells. If positioning of E6AP on Rpn10 (vs. ubiquitination of substrates by free E6AP or E6AP bound to some other portion of the proteasome) were truly what confers this privilege, then the E6AP knockdown should have little or no effect in the Rpn10 Δ RAZUL cells. Similarly, does overproduction of the RAZUL peptide as in Fig. 3b reduce ubiquitination on the proteasome?

We thank the reviewers for these recommendations. Indeed, when we knocked down E6AP in the hRpn10 Δ RAZUL cells, we did not observe any change in the amount of ubiquitin associating with the proteasome as tested by Rpt3 IP (Response Figure 2). One caveat with this approach, however, is that the Δ RAZUL cells exhibit a proteasomal defect in that the reduced levels of hRpn10 result in reduced association of lid components (see Rpn11 immunoprobe in Response Figure 2), as we have described in the new main Figure 3a where we demonstrate that boosting levels of hRpn10 in Δ RAZUL cells, including a truncated hRpn10 construct without the RAZUL domain, rescues the proteasome assembly defect caused by hRpn10 reduction. Therefore, we appreciate the Reviewer's further suggestion to overexpress the RAZUL peptide as an alternate approach. Upon expression of the RAZUL peptide, we did observe reduced levels of ubiquitinated proteins associating with the proteasome (Response Figure 3); this result is now included in the manuscript as Figure 4d.

Response Figure 2. Knockdown of E6AP does not affect levels of ubiquitinated substrates associating with the proteasome in Δ RAZUL cells. HCT116 or Δ RAZUL clone 13 cells were transfected with E6AP siRNA and subjected to Rpt3 IP. Lysates or immunoprecipitates were immunoprobed with the indicated antibodies.

Response Figure 3. RAZUL expression results in reduced ubiquitin levels associating with the proteasome. HCT116 cells were transfected with empty vector or myc-RAZUL and subjected to immunoprecipitation with IgG or Rpt3 antibodies. Lysates or immunoprecipitates were immunoprobed with the antibodies indicated to the left of the image.

2. Considering that Rpn10 is likely the major ubiquitin receptor on the proteasome, another potentially trivial explanation for the reduced ubiquitin associated with the proteasome in E6AP knockdown cells (Fig. 3) is that interaction with E6AP alters the affinity of Rpn10 for ubiquitinated substrates in the context of the proteasome. Is the interaction between proteasome-associated Rpn10 and ubiquitin/substrates similar when E6AP is knocked down?

In response to this concern, we tested directly through an *in vitro* pull-down assay whether E6AP binding leads to a reduction of affinity for ubiquitin chains by the 26S proteasome. The presence of E6AP did not affect binding by the proteasome to ubiquitin chains. We have now included this new data as Figure 4e in the revised text and below as Response Figure 4.

Response Figure 4. Interaction with E6AP does not reduce proteasome affinity for ubiquitin chains. **(a)** SDS-PAGE analyses for commercial E6AP and human erythrocyte 26S proteasome visualization by Coomassie staining (top panel) and immunoprobining for E6AP (bottom panel). **(b)** Pull-down assay for a commercially available mixture of His₆-tagged, non-cleavable K48-linked Ub₂/Ub₄ with incubation of human 26S proteasome (lane 6), 26S proteasome with equimolar E6AP (lane 7), or just E6AP (lane 8). E6AP or 26S proteasome was added to Ni-NTA agarose resin as negative controls (lane 4 and 5). K48-linked Ub₂/Ub₄, E6AP and 26S proteasome were loaded directly in lanes 1-3, as indicated.

3. Is there any evidence that substrate turnover is impaired or otherwise altered by loss of E6AP at the proteasome?

While there is evidence in the literature to suggest that E6AP promotes proteolytic activity of the proteasome (Tomaic and Banks, 2015), we were unable to find evidence that this promotion is dependent on the direct binding of E6AP to hRpn10. To test this possibility, we expressed the myc-RAZUL peptide as in Fig. 3b (now 4b) to knock off E6AP from binding hRpn10 and treated the cells with 30 µg/mL cycloheximide to compare the degradation rates of a short-lived substrate (p27^{kip1}, previously studied as a short lived substrate in Randles, et al.) as well as a previously identified E6AP substrate (hHR23A). We were unable to detect any major changes between the empty vector and RAZUL-expressing conditions for these two proteins, however more extensive study would be required to determine if a subset of substrates is affected by E6AP association with the proteasome and/or whether such an effect is dependent on specific cellular conditions or germane to only certain cell types.

We can include this figure in the manuscript as supplemental information if the Reviewer prefers; however, we have not done so in the current version as we feel a more exhaustive study is required to understand which substrates may be affected by E6AP at the proteasome as well as relevant cellular conditions.

Response Figure 5. Cycloheximide chase experiment probing stability of p27^{kip1} and hHR23A without and with displacement of E6AP from hRpn10 by RAZUL domain expression. Myc-hRpn10 RAZUL or empty vector (EV) was expressed in HCT116 cells and treated with 30 μ g/mL cycloheximide for the indicated timepoints. Lysates were immunoprobed with the indicated antibodies.

Tomic V, and Banks L (2015) Angelman syndrome-associated ubiquitin ligase UBE3A/E6AP mutants interfere with the proteolytic activity of the proteasome. *Cell Death Dis.* 2015 Jan 29;6:e1625. doi: 10.1038/cddis.2014.572.

Randles L, Anchoori RK, Roden RB, and Walters KJ. (2016) The Proteasome Ubiquitin Receptor hRpn13 and Its Interacting Deubiquitinating Enzyme Uch37 Are Required for Proper Cell Cycle Progression. *J Biol Chem.* Apr 15;291(16):8773-83. doi: 10.1074/jbc.M115.694588. PMID: 26907685

Minor points

1. As written, the abstract implies that Rpn10 Δ RAZUL cells display reduced ubiquitin associated with the proteasome. This was not directly tested (although see major point above).

Thank you for alerting us to this miscommunication; in trying to fit the abstract to 150 words we lost some meaning. We have now corrected this error.

2. p. 7: several references to figures are incorrect in the first two paragraphs.

Thank you; we have now fixed these typos.

3. Figure 3c is mislabeled as “d.”

Thank you for pointing this error out; we have now fixed this typo.

4. The authors indicate early in the manuscript that they have studied E6AP isoform 2; some (brief) commentary on whether the other isoforms of E6AP are likely to interact with Rpn10 in the same manner would be useful for naïve readers.

Excellent suggestion; we made this change in the revised text and included a supplemental figure (2b) for clarity.

5. The sentence in line 206 contains a grammatical error (replace contributes with recruits?).

Excellent suggestion; we made this change in the revised text.

REVIEWERS' COMMENTS:

Reviewer #1 (Remarks to the Author):

The authors have answered my concerns, and the structure validation report indicates a high quality NMR structure.

Reviewer #2 (Remarks to the Author):

In this revision of the manuscript from the Walters group entitled "Structure of E3 ligase E6AP with a novel proteasome-binding site provided by substrate receptor hRpn10," the authors have added additional experimental data and included edits to address initial concerns of two reviewers. This Reviewer's suggested edits for the initial submission centered on strengthening the conclusion that changes in the proteasome-bound polyubiquitin conjugates in Rpn10deltaRAZUL cells were truly due to loss of E6AP recruitment rather than other trivial explanations, such as altered affinity of hRpn10deltaRAZUL for ubiquitin conjugates in the context of the proteasome, or alterations in the affinity of hRpn10deltaRAZUL for the proteasome itself.

In this revised manuscript, the authors have provided robust evidence that these trivial explanations are not responsible, which in conjunction with the initial evidence provided, strongly supports their conclusions. The authors have in my opinion satisfactorily addressed all of my initial concerns.